# Rosemary essential oil and its components 1,8-cineole and α-pinene induce ROS-dependent lethality and ROS-independent virulence inhibition in *Candida albicans*

**Zinnat Shahina**[1], **Raymond Al Homsi**[2☯], **Jared D. W. Price**[1☯], **Malcolm Whiteway**[2], **Taranum Sultana**[1]*, **Tanya E. S. Dahms**[1]*

**1** Department of Chemistry and Biochemistry, University of Regina, Regina, Saskatchewan, Canada,
**2** Centre for Structural and Functional Genomics, Concordia University, Montreal, Quebec, Canada

☯ These authors contributed equally to this work.
* Tanya.Dahms@uregina.ca (TESD); taranum.sultana0@gmail.com (TS)

**Data Availability Statement:** All relevant data are within the paper and its Supporting information files.

## Abstract

The essential oil from *Rosmarinus officinalis* L., a composite mixture of plant-derived secondary metabolites, exhibits antifungal activity against virulent candidal species. Here we report the impact of rosemary oil and two of its components, the monoterpene α-pinene and the monoterpenoid 1,8-cineole, against *Candida albicans*, which induce ROS-dependent cell death at high concentrations and inhibit hyphal morphogenesis and biofilm formation at lower concentrations. The minimum inhibitory concentrations (100% inhibition) for both rosemary oil and 1,8-cineole were 4500 µg/ml and 3125 µg/ml for α-pinene, with the two components exhibiting partial synergy (FICI = 0.55 ± 0.07). At MIC and 1/2 MIC, rosemary oil and its components induced a generalized cell wall stress response, causing damage to cellular and organelle membranes, along with elevated chitin production and increased cell surface adhesion and elasticity, leading to complete vacuolar segregation, mitochondrial depolarization, elevated reactive oxygen species, microtubule dysfunction, and cell cycle arrest mainly at the G1/S phase, consequently triggering cell death. Interestingly, the same oils at lower fractional MIC (1/8-1/4) inhibited virulence traits, including reduction of mycelium (up to 2-fold) and biofilm (up to 4-fold) formation, through a ROS-independent mechanism.

## Introduction

*Candida albicans* (*C. albicans*) is a commensal organism that acts as an opportunistic pathogen, in particular in immunocompromised patients [1, 2] such as those with HIV, or undergoing chemotherapy and organ transplantation [2, 3]. Clinically used antifungals suffer from low bioavailability, poor gastrointestinal absorption, and combined with developing resistance in *C. albicans* strains warrants a search for alternative antifungals [4].

**Funding:** This work was supported by Natural Science and Engineering Research Council Discovery Grant (NSERC DG; RGPIN-2018-06649), Saskatchewan Health Research Foundation Collaborative Innovation Development and Canada Foundation for Innovation (CFI) grants to TESD, a NSERC DG (RGPIN-2017-4799) and Canada Research Chair (950-228957) to MW. TS was partially supported by the Faculty of Science and CFI infrastructure operating fund to TESD. ZS was partially supported by the Faculty of Graduate Studies and Research at the University of Regina. There was no additional external funding received for this study.

**Competing interests:** The authors have declared that no competing interests exist.

Rosemary (RM) oil, extracted from a member of the *Lamiaceae* family *Rosmarinus officinalis* L., is rich in the 1,8-cineole (27–37%) and α-pinene (8–20%) [5–9], the ratio of which varies as a function of geographical source. Extensive data exist on the chemical constituents, and on antimicrobial and antibiofilm properties of RM oil against *Candida* spp. including *C. albicans*, but a detailed antifungal mechanism remains elusive [10, 11]. Bakkali et. al. [12] report that the biological activities of most essential oils (EOs) are primarily linked to their major constituents. Eucalyptus (*Eucalyptus intertexta*) and tea tree (*Melaleuca alternifolia*) oils [13] also contain 1,8-cineole [14], which inhibit biofilm formation [15] and destabilize the *C. albicans* cytoplasmic membrane, increasing membrane permeability, allowing the essential oil to penetrate organelle membranes [16], and induce oxidative damage to DNA and subsequently trigger apoptosis [17, 18]. Further, (+)-α-pinene perturbs cell integrity, and inhibits respiration and ion transport processes [19].

Chemogenomic profiling with the budding yeast *Saccharomyces cerevisiae* has been used extensively to identify targets of many bioactive compounds [20–22], and this approach has been more recently applied to other fungi [23–25]. The azoles, echinocandins and pyridine amide classes of antifungals have been shown to target essential gene products Erg11 [26], Fks1 [27] and the biosynthesis of GPI anchors through inhibition of Gwt1 [28]. In *S. cerevisiae*, thymol inhibits transcription of *EST2* and therefore telomerase [29], and eugenol targets two permeases, Tat1p and Gap1p, involved in aromatic and branched-chain amino acid transport [30]. In *C. albicans* carvacrol targets tryptophan biosynthesis via hypersensitivity to *trp1*, *trp2*, *trp3*, *trp4*, *aro1*, and *aro2* [16] and geraniol modulates CaCdr1p efflux pump activity [31].

Chemogenomics profiling is a powerful method to gain insight into potential modes of drug action, but such insights must be validated by other methods, for example biochemical and microscopic approaches, including laser scanning confocal (LSCM) and atomic force microscopy (AFM) [32]. Fluorescence-based optical microscopy, including LSCM, enables the visualization of molecules within live samples in real time [33, 34], while AFM can probe the cell surface on the nano scale along with cell stiffness, surface viscoelasticity and molecular interactions [35]. In this study, these microscopic methods provide insight into cell morphogenesis [36, 37], distribution of nucleic acids, protein (Tub2) location [38], cell cycle arrest [39], ROS accumulation [40] and cell death/apoptosis [41, 42].

In this study we combine chemogenomics, microscopy and bioassays to examine mechanistically how sublethal doses of RM oil and its two major components, 1,8-cineole and α-pinene, cause cell death and impact *C. albicans* virulence factors.

## Material and methods

### Strains, media and growth conditions

*C. albicans* RSY150, ATCC10231 (clinical reference), clinical and mutant strains, used in this study are described in the supplementary data (S1 Table) and were kind gifts of Dr. Richard J. Bennett (Department of Molecular Microbiology and Immunology, Brown University, USA), Dr. Lois L. Hoyer (Department of Veterinary Pathobiology, University of Illinois at Urbana-Champaign, Urbana, USA) and Dr. Jessica Minion (Department of Microbiology, Regina Qu'Appelle Health Region, Regina, SK Canada), respectively. All *C. albicans* strains were stored as 50% glycerol stocks at −80°C and were freshly revived on yeast peptone dextrose (YPD) media containing 1% yeast extract, 2% peptone, 2% glucose and 2% bacto-agar prior to each experiment. Starter cultures were grown from single colonies were grown with agitation (200 rpm) at 30°C in YPD broth for 16–20 h. Fetal bovine serum (10%) was used for hyphal induction and biofilm formation in YPD media, and spider media (1% peptone, 1% yeast extract, 1% manitol, 0.5% NaCl, and 0.2% $K_2HPO_4$) was used for mycelial growth at 37°C.

## Essential oils and positive controls

Steam distilled essential oil from *Rosmarinus officinalis* L. (whole plant) and its two components (EOCs) 1,8-cineole (99%) and (+)−α-pinene ($\geq$ 99%) were purchased from Aroma force (CAT# 6658117001 Canada), Acros organics (CAT# 110345000, New Jersey, USA) and Sigma Aldrich (CAT# 268070, St. Louis, MO, USA), respectively. Bacto™ agar, yeast extract and peptone were obtained from Difco (BD Biosciences, NJ, USA), and all other chemicals were purchased from Sigma Chemical Co. (St. Louis, MO, USA). Different concentrations of polyoxyethylenesorbitan monooleate (Polysorbate 80, TWEEN® 80, CAT # P1754, Sigma-Aldrich, St. Louis, MO, USA) were tested, with 0.2% TWEEN® 80 found effective for EO(C) solubility and no impact on fungal growth. Amphotericin B (Amp B), $H_2O_2$ and nocodazole, (CAT# A9528, H1009 and M1404, respectively, Sigma-Aldrich, St. Louis, MO, USA) served as positive controls for various assays, with media only as the negative control.

## GC-MS analysis

GC-FID and GC-MS were performed with an Agilent 7890A GC according to previously reported methods [43, 44]. Briefly, 1 µl of dilute RM oil (1:10 in ethanol) was injected onto a HP-5 column ($30 \times 0.32$ mm $\times 0.25$ µm) and separated using a helium carrier gas (constant flow of 1.1 mL/min), with the following temperature gradient: 60˚C for 5 min, increased to 210˚C at 3˚C/min and to 260˚C at 10˚C/ min. The GC-FID injector and detector temperatures were 250˚C and the split ratio of injection was 20:1. For GC-MS under similar conditions, 0.2 µl sample was injected with a split ratio of 200:1 and separated on a HP-5MS column ($30 \times 0.25$ mm $\times 0.25$ um). Retention indices (RI) were determined using a C8–20 standard (~ 40 mg/L each, in hexanes) for both methods, averaged for each oil component and identified using the NIST14 database and reported literature [43, 45].

## *Candida* culture preparation

An overnight culture of *Candida* at log phase were diluted with YPD broth to a final $OD_{600}$ of 0.001, which is equivalent to $1.2 \times 10^5$ cells/ml or to 0.5 McFarland standard for both the MIC and checkerboard assays as per the CLSI (Clinical and Laboratory Standards Institute) guidelines for yeast [46].

For all other biochemical assays, cells were harvested from mid log phase ($OD_{600}$ ~ 0.6–0.8) after 12–14 h incubation, giving more metabolically active cells, and diluted to ~$10^7$ CFU/mL prior essential oil exposure, as described in Shahina et al. [47]. Following EO(C) exposure (4 h unless otherwise stated), samples were normalized to $10^5$ cells/mL (optimal, unsaturated signal) and assayed in 12, 24 or 96 microwell plates, depending on assay type. For microscopic imaging, a cell density of ~$10^7$ CFU/mL was ideal to ensure sufficient coverage.

## Minimum inhibitory concentration assay

The MIC of *C. albicans* strains (S1 Table) exposed to EO(C) working solutions (S2 Table) were determined as the lowest concentration of test entity that causes 100% decrease in optical density ($OD_{600}$) compared with the control, in accordance with CLSI guidelines [46] as described in Shahina et al. [47].

## FICI determination by checkerboard assay

The antimicrobial activity of the EOCs in combination (1,8-cineole/α-pinene) were assessed by checkerboard assay, with slight modification from literature protocols [48]. Serial dilutions (2.22-fold) of the EOC pair were prepared in YPD/TWEEN® 80 using similar concentration

ranges as for the MIC assays. A 90 μl volume of one EOC was added to the rows and the second component to columns of a 96-well microtiter plate in decreasing concentrations. A 20 μl yeast inoculate ($1.2 \times 10^5$ cells/mL) was added to each test well, along with two growth controls in YPD/TWEEN® 80, negative controls containing media only, and blank with EOC in YPD/TWEEN® 80 without yeast inoculate. The plates were then incubated overnight at 30˚C and the results of fractional inhibitory concentration (FIC) and FIC index (FICI) calculated from the following formulae 1–2:

$$FIC = \frac{MIC \text{ in combination}}{MIC \text{ alone}}; \qquad (1)$$

$$FICI = FIC1 + FIC2; \qquad (2)$$

The FICI was interpreted as synergistic (FICI ≤ 0.5), partially synergistic (0.5 < FICI ≤ 0.75), additive (0.76 < FICI ≤ 1), indifferent (1 < FICI ≤ 4.0) or antagonistic (FICI > 4.0) [49]. Isobologram was constructed by plotting the concentration of one EOC against the other, with a straight line representing zero interaction, a concave curve indicating synergistic (FICI ≤ 0.5) and additive (0.5 < FICI < 1) interactions and a convex curve representing antagonistic interactions (FICI > 4) [50]. In this study the microscopic examination used fractional (1/8–1/2) lowest combined lethal concentrations of the two components which we refer to as FICI, 1/2 FICI and 1/4 FICI.

## Identification of sensitive and resistance strains by chemogenomic profiling

The GRACE 1.0 library strains were tested for sensitivity or resistance to specific EOCs. Cells were inoculated using the pinning tool robot into YPD media in a 96-well microtiter plate, which was expanded into a 384 well plate by transferring 25 μL from each 96-well plate into 4 wells of the 384-well plate. An equal aliquot (25 μL) of 1,8-cineole (3.5%) and α-pinene (1.1%) stock solutions in YPD with 0.2% Tween 80 were transferred into each well of a 384-well plate by a liquid handling Biomek FX$^P$ robot (Beckman Coulter Canada, Mississauga, ON, Canada), the plate sealed with parafilm and incubated for 24 h at 30˚C with shaking (220 rpm), corresponding to. The ratio of [$OD_{600}$ treated] to [$OD_{600}$ YPD] was used to remedy errors associated with slow or elevated growth rates in the mutants. Endpoint readings were set as the antifungal concentrations causing at least 90% growth inhibition after 3d incubation compared to growth of the control (Cai4), to determine EOC impact.

A protein-protein interaction network with resistant and sensitive interactions was constructed from the experimental data using the online STRING (http://string-db.org/) database [51] and visualized using Cytoscape v 3.7.0 (https://cytoscape.org) [52]. Protein network interactions were further grouped into nodes of clusters using the MCODE application (https://bio.tools/mcode), which groups molecules based on pre-specified criteria. The localization of *C. albicans* resistant and sensitive genes were determined using the Gene Ontology Term Finder tool [53] of the *Candida* Genome Database (CGD) (www.candidagenome.org).

## Microscopy

For optical imaging, a 5 μL aliquot of a treated or untreated *C. albicans* suspension in PBS was pipetted onto glass microscope slides (sterilized with 70% ethanol), then covered with a clean (methanol/acetone) coverslip and sealed with nail polish. Coverslips for AFM imaging were cleaned according to Paul et al. [54].

a. Optical imaging in bright-field was used to assess candidal morphology and biofilms using a stereomicroscope (Nikon's SMZ 1500, Japan) with digital image capture, and gross morphological changes (budding yeast, pseudohyphae and hyphae) identified using the transmitted light configuration of an epifluorescence microscope (Carl Zeiss Axio Observer Z1 inverted microscope, Oberkochen, Germany). Fluorescently labeled molecules were localized within different organelles of live or fixed cells in epifluorescence mode (Axio Observer Z1) and using a laser scanning confocal microscope (LSCM; Carl Zeiss LSCM 780, Oberkochen, Germany) at 63×.

b. AFM in quantitative imaging (QI™) mode with a Nanowizard III AFM (JPK Instruments, Berlin, Germany) was used to assess cell surface mechanics and topography at ultra-high resolution in response to EO(C)s. Briefly, suspensions of *C. albicans* RSY150 overnight cultures treated with RM oil for 24 h, were washed three times with PBS and deposited onto poly-L-Lysine coated cover slips for 1 h, fixed with formalin and air dried prior to AFM imaging. Samples were imaged using silicon nitride cantilevers (HYDRA6R-200NG; Nanosensors, Neuchatel, Switzerland) with calibrated spring constants ranging from 0.03 to 0.062 N/m. QI™ force curves (JPK software, SPM version 5.1, JPK Instruments, Berlin, Germany) for each pixel of a $128 \times 128$ raster scan were collected using a Z-length of 7 μm and a raster scan of 100 μm/s. A subset (20) of force curves within a $200 \times 200$ nm square in the center of the cell were collected from each biological replicate (3), batch processed (JPK software), and exported (Excel). Adhesion was determined using the distance between the lowest point and baseline of the retract curve, and Young's moduli, a measure of cell envelope elasticity, estimated using the Hertz model (JPK software) and the approach curve. A square at the cell mid-point from QI™ height images was used to calculate surface roughness using the JPK software [55].

## Morphological analyses

Changes to *C. albicans* morphology following EO(C) exposure were determined using calcofluor white (CFW) staining as described in Shahina et. al. [47].

## Membrane integrity assays

a. The membrane depolarization assay was used to determine the impact of EO(C)s on the cytoplasmic membrane with the membrane potential-sensitive probe, 3,3′-diethylthiadicarbocyanineiodide [Dis-C2(3)] (Sigma-Aldrich, St. Louis, MO, USA), as previously described [56, 57]. Briefly, approximately $1 \times 10^7$ CFU/mL of mid logarithmic phase *C. albicans* RSY150 and ATCC10231 cultures were exposed to EO(C)s at MIC, 1/2,1/4,1/8 and 1/16 MIC for 4 h and at their FICI in a 96 well plate in YPD media, along with cultures treated with Amp B at MIC, or reagents only as blank. The dye dissolved in DMSO was added 5 min prior to the addition of test compounds, to a final concentration of 2 μM (dye) and 1% (DMSO). Changes in fluorescence intensity were measured before and after 1 to 4 h incubation at 30 ˚C using a microplate reader ($\lambda_{ex} = 560$ nm; $\lambda_{em} = 580$ nm) and further examined by LSCM ($\lambda_{ex} = 560$ nm; $\lambda_{em} = 580$ nm).

b. Membrane damage of *C. albicans* RBY1132 (parent strain of RSY 150) was assessed using the method described by Setiawati et al. [58], with slight modification. Briefly, following 4 h exposure of a *Candida* suspension in a 24 well plate to EO(C)s, $H_2O_2$ or Amp B at MIC and 1/2 MIC, cell density was normalized to $10^5$ cells/mL and 200 μL of the suspension transferred to wells of a flat-bottom 96 well microplate, followed by the addition of propidium iodide (2 μL; 1 μg/mL), and incubation for 30 min at 30˚C in the dark. Lastly, epifluorescence

($\lambda_{ex}$ = 493 nm; $\lambda_{em}$ = 636 nm) was measured, and the fluorescence intensity quantified (ZEN version 2.3 software) from 100 cells per biological replicate (3).

## Identification of vacuolar defects

Following a 4 h exposure to EO(C)s and Amp B at MIC and 1/2 MIC, vacuolar segregation in *C. albicans* RYS150 and ATCC10231 was imaged using the transmission configuration of the epifluorescence microscope at 63× from a total of 10 different fields per biological replicate (3), from which 300 individual cells were counted and plotted.

## Determination of mitochondrial health

The impact of EO(C)s on mitochondria was determined using published protocols [59]. Following a 4 h EO(C) exposure, *C. albicans* were harvested, washed and resuspended in PBS to $10^5$ cells/mL, 100 μL of the suspension transferred to wells of a 96 well microplate, incubated with 5 μl of MitoTracker® Deep Red (Invitrogen) (100 nM) for 30 min at 30˚C in the dark, and imaged by LSCM ($\lambda_{ex}$ = 644 nm; $\lambda_{em}$ = 665 nm). Poor mitochondrial condition was identified as cells with limited fluorescence, and intensities quantified (ZEN software) from 100 cells per biological replicate (3).

## Quantification of ROS generation

The effect of EO(C)s on the generation of intracellular ROS in *C. albicans* RBY1132 was assessed using methods adapted from published protocols [60]. Following 4 h oil exposure to EO(C)s (MIC—1/16 MIC) or the positive control 25 mM $H_2O_2$ [61], 10 μmol/L of the ROS-sensitive probe 2,7- Dichlorodihydrofluroscein diacetate (DCFDA) was added and incubated for 30 min. Cells were harvested, washed twice with PBS, suspended in the same buffer to $10^5$ cells/mL and 100 μL of the suspension transferred to each well (96) of a flat-bottom microplate. Fluorescence intensity ($\lambda_{ex}$ = 485 nm; $\lambda_{em}$ = 528 nm), indicating ROS levels, was recorded using a BioTek Synergy HTX multi-mode microplate reader (Northern Vermont, USA) or viewed by epifluorescence microscopy ($\lambda_{ex}$ = 485 nm; $\lambda_{em}$ = 528 nm).

## Cell cycle and microtubule arrangement

To determine the impact of EO(C)s on cell cycle and MTs at mid logarithmic phase, *C. albicans* RSY150 cells expressing Tub2-GFP and Htb-RFP were treated (4 h) with EO(C)s at MIC and 1/2 MIC [47, 62], and imaged by LSCM to determine the expression tagged β-tubulin (Tub2-GFP) using an argon ion laser ($\lambda_{ex}$ = 488 nm; $\lambda_{em}$ = 512 nm) and tagged histone protein B (Htb-RFP) with the HeNe laser ($\lambda_{ex}$ = 543 nm; $\lambda_{em}$ = 605 nm). Cell cycle phase was identified from microscopic images based on the presence or absence of a visible bud, the size of the bud relative to the mother cell, and nuclear organization (Htb-RFP). Tub2-GFP incorporation into MTs (long, short) or lack of MT formation (diffuse) were identified by counting and measuring the intensity and supermolecular length of green fluorescent regions [63] using ImageJ (http://rsb.info.nih.gov/ij/) from 100 cells per biological replicate (3). The ZEN software was used to manually count cell cycle or MT structures for 250 cells from each biological replicate. *C. albicans* strains incubated with media only or nocodazole were used as negative and positive controls, respectively.

## *C. albicans* morphological transition/virulence inhibition assays

**(a) Serum-induced germ tube formation assays.** The germ tube assay followed the literature [64], with slight modification. Briefly, a yeast suspension of $(1 \times 10^7$ CFU/mL) was

prepared from mid logarithmic phase cells in pre-warmed YPD with 10% fetal bovine serum (FBS) and deposited into a 24 well plate with EO(C)s at MIC and 1/2 MIC, component pairs at FICI and 1/2 FICI, or the positive control Amp B. *C. albicans* strains incubated with YPD in 10% FBS served as negative controls. Following 4 h incubation, microscopic slides were prepared with an aliquot of *C. albicans* (RSY150, ATCC10231, Cli-1,Cli-2, Cli-3) culture stained with CFW, and the proportion of germ tubes, hyphal or pseudohyphal forms evaluated by epifluorescence imaging. Results are expressed as the average number of germ tube forming cells/ 100 cells from each biological replicate (3).

To determine how long germ tube inhibition took at FICI (1,8-cineole + α-pinene), six different time points were assessed according to previous methods [64]. The proportion of normal and germ tube forming cells were evaluated in each culture by epifluorescence at 4, 8, 12, 16, 20 and 24 h.

**(b) Mycelial growth assay.** Mycelial growth was assessed on spider media agar plates with or without (negative control) EO(C)s at various concentrations (1/2 MIC to 1/16 MIC), or the respective component pairs at their FICI. Aliquots (2 μL) of a *C. albicans* RSY150 cell suspension ($1 \times 10^7$ CFU/mL) at mid logarithmic phase were spotted onto spider agar media (CAT# 83.3921.500; Sarstedt, Nümbrecht, Germany) according to published protocols [64–66]. The plates were incubated for 6 days at 37 ˚C and the presence of colonies and hyphal growth at the colony edges imaged by stereomicroscopy at 2× and 4× magnification.

Inhibition of morphological switching was assessed for *C. albicans* pre-treated with EO (C)s according to published protocols [64–66], with slight modification. Briefly, five strains of *C. albicans* (RSY150, ATCC10231, Cli-1, Cli-2 and Cli-3) were treated with EO(C)s at 1/2 MIC and the two components at 1/2 FICI for 4 h. Following incubation and three washes with PBS, 2 μL aliquots of treated cells ($1 \times 10^5$ cells/mL) were spotted onto individual 24 wells plate, containing solid spider medium prepared without EO(C)s. The plates were incubated for 6 days at 37 ˚C and the morphology of the fungal colony imaged in bright-field on a stereomicroscope and photographed using a digital camera. This experiment was conducted in quadruplicate.

**(c) *In vitro* biofilm inhibition assay.** To evaluate biofilm inhibition, a 100 μl aliquot of $1 \times 10^7$ CFU/mL *C. albicans* RSY150 suspension in YPD with 10% FBS was inoculated into individual wells of 96-well polystyrene tissue culture plates (flat bottom; Sarstedt, Nümbrecht, Germany). To allow initial adhesion, plates were incubated for 90 min at 37 ºC in an incubator-shaker at 75 rpm [66]. The biofilms were allowed to develop for 24 h, treated with each of the EO(C)s and evaluated as described below.

The MTT assay was used to determine the metabolically active *Candida* inhabiting the biofilm population as per Tsang et al. [66], but with slight modification. A stock solution (5 mg/ml) of MTT (3-(4,5-dimethylthiazol-2-yl)-2,5-diphenyl tetrazolium bromide) was prepared by dissolving MTT powder (Sigma Aldrich) in PBS followed by filtration (0.22 μm pore size, sterile). *C. albicans* RSY150 were prepared according to the previous section, washed twice with 200 μl PBS to remove non-adherent cells, and fresh medium (100 μl, YPD in 10% FBS) either with or without EO(C)s, followed by incubation at 37˚C for 24 h. Wells containing only *C. albicans* in YPD broth with 10% FBS were used as negative controls. Following incubation, the supernatant was aspirated, the MTT solution (50 μl, 1 mg/mL working solution) added, incubated for 4 h at 37˚C, MTT solution removed with gentle aspiration, 100 μl DMSO added to dissolve the dark blue formazan crystals [67], and the plates further incubated for 15 min at room temperature. Absorbance was determined by microplate reader at 570 nm (Biotek Epoch; Northern Vermont, USA) and the percentage of

biofilm formation inhibition calculated according to Eq 3 [68]:

$$\% \text{ Inhibition } = 100 - \left[ \left\{ \frac{A_{570\text{ nm}} \text{ EO(C)}}{A_{570\text{ nm}} \text{ control}} \right\} \times 100 \right]; \quad\quad (3)$$

The crystal violet (CV) assay was used to quantify biofilm formation for *Candida* pretreated with EO(C)s at 1/2 MIC following the published protocol [69], with some modification. Five *C. albicans* strains (RSY150, ATCC10231, and three clinical isolates) were incubated with EO (C)s at 1/2 MIC, with untreated *Candida* and those exposed to Amp B serving as a negative and positive controls, respectively, and wells containing only YPD in 10% FBS as blanks. Cells were then washed three times with PBS to avoid a 'carryover' effect, washed with YPD broth to remove PBS, and resuspended in fresh YPD with 10% FBS. A 100 μl aliquot ($1 \times 10^5$ cells/mL) of the treated suspension was transferred into each well of 96 well plate, and incubated at 37°C in an incubator-shaker at 75 rpm for 24 h. Following biofilm formation and growth, each well was washed twice with 150 μl sterile PBS buffer, the plate dried for 20 min at 37°C, and stained and incubated with 100 μl of 1% aqueous CV solution (Sigma Aldrich) for 20 min at 37°C. The stained biofilms were washed three times with 200 μl sterile, ultrapure (Milli-Q) water and destained with 200 μl of 95% ethanol for 45 min. A 100 μl aliquot of the resulting solution was transferred to a new plate with 100 μl of destaining solution, absorbance (595 nm) measured on a plate reader (Biotek Epoch; Northern Vermont, USA) followed by blank subtraction. Each strain was tested in triplicate in the same plate and biofilm production quantities reported as the arithmetic mean of absorbance values from four biological replicates.

**(d) Biofilm imaging.**   Biofilms grown from *C. albicans* (500 μl) in 24 well tissue culture plates (Cat # 83.3922, Sarstedt, Nümbrecht, Germany) prepared as above were imaged with a stereomicroscope at 4×, to determine the presence or absence of biofilm, compared with a blank well consisting of equal volumes of experimental media lacking *Candida*.

## Statistical analyses

Statistical analyses were conducted using GraphPad Prism (Version 6.0; La Jolla, CA, USA). Unpaired Student *t*-tests with a Welch's correction at 95% confidence interval were used to determine if two data sets were the same with respect to a tested variable. One-way ANOVA, followed by Dunnett's multiple post-test, was used to compare all data versus control for ungrouped data with more than two variables. The extent of statistical significance is indicated by asterisks: * ($p < 0.05$), ** ($p < 0.01$), *** ($p < 0.001$), **** ($p < 0.0001$), with the absence thereof indicates no statistical significance ($p > 0.05$) and graphical error bars indicate the standard error mean (SEM) from at least three independent biological replicates. The "r" value, indicating linearity, was calculated in Excel using the built-in Pearson correlation coefficient function.

## Results

### 1,8-cineole and α-pinene are major components of rosemary oil

The components (S3 Table) of rosemary (RM) oil analyzed by gas chromatography-flame ionization detection (GC-FID) and gas chromatography–mass spectrometry (GC–MS) were identified using previously reported retention indices (RI) [43, 45]. Twenty compounds were identified, with 1,8-cineole as the major component (53%, S1 Fig), α-pinene as the next most abundant (12%), along with other components such as camphor (8.6%), β-pinene (7.3%), β-caryophyllene (4.4%), linalool (0.6%) camphene (4%), p-cymene (1.8%) and α-terpineol (1.7%).

## Clinical *C. albicans* strains have different susceptibilities to RM EO(C)s

The broth microdilution method was used to examine the impact of RM, 1,8-cineole and α-pinene oils on the planktonic growth of *C. albicans* strains. Lab strains RBY1132 and RSY150 showed RM oil and 1,8-cineole had minimum inhibitory concentrations (MICs) of approximately 4500 μg/ml, whereas ATCC10231 and clinical strains isolated from genital and blood samples were more resistant (7000 and 9250 μg/ml, respectively). Out of the seven strains (S2A Fig), five were susceptible to α-pinene with a MIC of 3125 μg/ml, except one genital and one blood strain which both exhibited higher MIC values (4250 μg/ml). RSY150 and ATCC10231 treated with amphotericin B (Amp B) and nocodazole had MIC values of 0.31 and 20 μg/ml, respectively (S2A Fig).

## The major components of RM oil are partially synergistic against *C. albicans*

The FIC (Fractional inhibitory concentration) index of 0.55 ± 0.07 for *C. albicans* RSY150 exposed to 1,8-cineole and α-pinene indicates partial synergy (FICI > 0.5 to ≤ 1), seemingly at odds 1,8-cineole and RM having similar MICs, but may represent antagonism of other minor components. The checkerboard titer assay (S2B Fig) revealed the MIC of 1,8-cineole alone to be 4205 μg/ml and significantly lower (1911 μg/ml) in the presence of α-pinene at 293 μg/ml, which alone had a MIC of 3125 μg/ml, consistent with the concave isobologram (S2C Fig).

**(i) ROS-dependent anti-candida activity of EO(C)s at high concentration.** With exposure to higher levels (1/2 MIC–MIC) of EO(C) concentrations, *C. albicans* cell death follows a ROS-dependent pathway.

## Chemogenomic profiling of *C. albicans* exposed to RM EO(C)s

Despite exhaustive chemogenomic profiles for *C. albicans* and traditional antifungals, there are limited studies focusing on EOs and their components [16, 29–31]. We analyzed the GRACE 1.0 library [23] for response to cineole and pinene. There were 40 and 50 strains resistant to 1,8-cineole and α-pinene, respectively, with an overlap of 5 strains (Fig 1A), including strains with mutations in *UBX7* (Putative UBX-domain protein), *YOL092W* (Possible G-protein coupled receptor; vacuolar membrane transporter for cationic amino acids), *ECM14* (putative metallocarboxypeptidase), *CaORF6_2253* (Protein of unknown function) and *PNG1* (Putative peptide N-glycanase). Of 32 and 37 strains sensitive to 1,8-cineole and α-pinene, respectively, there was an overlap of 8 strains (Fig 1B) with knock-outs of *PDS5* (predicted role in sister chromatid condensation and cohesion; cell-cycle regulation and periodic mRNA expression), orf19.1125 (predicted hydrolase; induced by nitric oxide), *HIP1* (putative histidine permease), *SNG1* (putative membrane transporter), *NCA2* (predicted role in aerobic respiration, mRNA metabolic process), *YIL110W* (putative protein-histidine N-methyltransferase), *PHA2* (putative prephenate dehydratase) and *YBR005W* (potential role in chitin localization, vesicle-mediated transport and ER membrane localization).

*PCK1*, *ADE5,7* and *GDB1* are involved in 1,8-cineole resistance and show network interactions, for which *ADE5*, encoding a bifunctional ribotide synthase, is the common interactor. On the other hand, a considerable number of genes conferring resistance to α-pinene interact (*GIT1*, *PHO84*, *PHM7*, *YAK1*, *PIL1*, *GLN3* and *ERF2*, *AKR1*, *GSG1*, *MON1*, *MDN1*, *VPS13*, *KEX2*, *ECM14*, *GLR1*, *XYL2*, *RKI1*, *NEP1*). Of interest are three proteins with highly interconnected subnetworks (Fig 1A and 1B): *ECM14*, encoding a component of the putative metallo-carboxypeptidase, *KEX2*, a protease involved in pheromone and killer toxin maturation, and *VPS13* encoding the vacuolar protein sorting-associated protein involved in recycling of

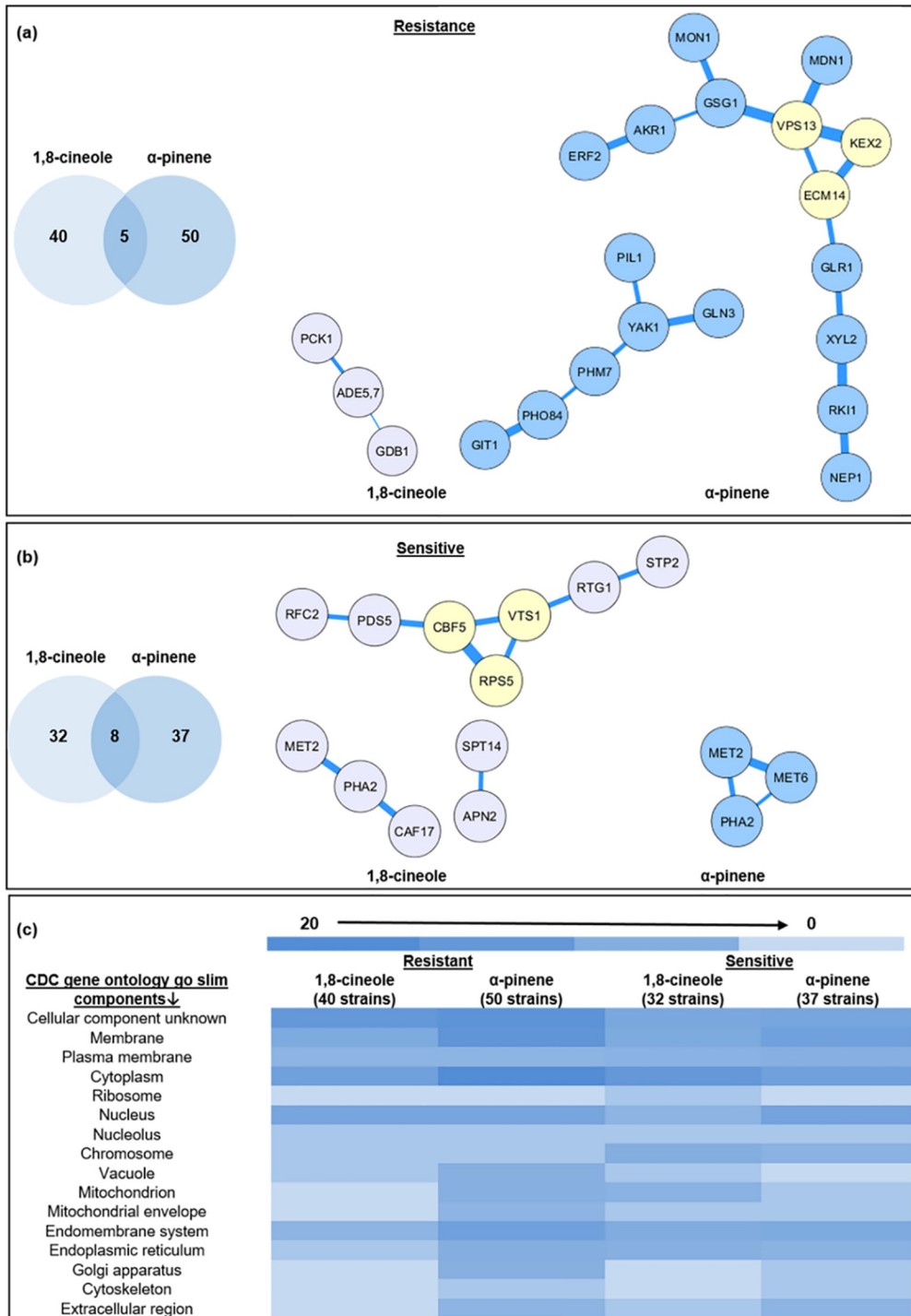

**Fig 1. Overview of PPI Network of *C. albicans* genes for strains resistant and sensitive to EOCs. (a,b)** Comparison of genes showing resistance and sensitivity to 1,8-cineole and α-pinene, including protein-protein interaction networks. **(a)** There were 5 genes conferring resistance to both 1,8-cineole and α-pinene and **(b)** 8 conferring sensitivity to both 1,8-cineole and α-pinene. Thick lines represent very high confidence, yellow nodes represent highly interconnected subnetworks. **(c)** Heatmap of gene product localization to cellular components.

membrane proteins in the endocytic compartment. Of the sensitive strains, there are three sets of interacting genes for 1,8-cineole (*MET2*, *PHA2*, *CAF17*; *APN2*, *SPT14* and *RFC2*, *PDS5*, *CBF5*, *RPS5*, *VTS1*, *RTG1*, *STP2*), with highly interconnected subnetworks between *CBF5*, *VTS1* and *RPS5*, which encode a ribonucleoprotein complex that catalyzes pseudouridylation of rRNA, a RNA-binding protein involved in post-transcriptional regulation through transcript degradation and ribosomal 40S subunit protein, respectively. Only 3 genes (*MET6*, *MET2* and *PHA2*) associated with α-pinene sensitivity interact, encoding 5-methyltetrahydropteroyltriglutamate-homocysteine methyltransferase, homoserine O-acetyltransferase and prephenate dehydratase (Fig 1B).

Functional analysis (S4 Table) shows that two genes involved in quinone binding (*AMO1*, *FESUR1*) confer resistance ($p < 0.05$) to 1,8-cineole, and two involved in carboxypeptidase activity (*CPY1*, *ECM14*) confer resistance to α-pinene. While there are two genes that conferred sensitivity to 1,8-cineole, neither have assigned functionality, and two (*C5_05040W_A*, *MNT2*) related to alpha-1,2-mannosyltransferase activity conferred α-pinene sensitivity. In terms of predicted cellular responses and localization (S3 Fig), gene products associated with membranes but not particularly favouring any given organelle were most commonly impacted (Fig 1C), indicative of a general stress response to membrane damage at MIC and higher fractional MICs.

## Chitin increase and altered cell wall nanomechanical properties with RM EO(C)s

*C. albicans* RSY150 exposed to RM oil or its components morphologically transformed into either swollen cells or chains of interconnected cells (Fig 2A), indicative of a separation defect, while the clinical reference strain (ATCC 10231) became aggregated (S4A Fig). Similar separation defects induced by fluconazole are directly linked to increased cell wall chitin content, associated as a general stress response [70]. Consistent with this, we found *C. albicans* RSY150 exposed to RM oil had significantly ($p < 0.01$ for 1,8-cineole, α-pinene, $p < 0.05$ for RM oil) higher CFW fluorescence intensity than controls (Fig 2A), indicating elevated chitin content. (Fig 2B), similar to that for ATCC10231 (S4B Fig). Amp B at MIC and the components (1,8-cineole and α-pinene) at their FICI also generated a significant ($p < 0.001$) increase in chitin levels (Fig 2B).

To determine if increased chitin content altered the cell surface ultrastructure and mechanical properties, *C. albicans* RSY150 cells exposed to 1/2 MIC RM oil were analyzed by QI™ AFM (Fig 2A and 2B) to generate images and $\geq 20607$ force curves. RSY150 (n = 17) exposed to RM oil at 1/2 MIC for 24 h had no change ($p < 0.3480$) in surface roughness (4.0 ± 1.1 nm) compared to control (3.6 ± 1.0 nm), but the cell surface adhesion (7.6 ± 0.5 nN) and wall elasticity (66.7 ± 9.6 MPa) were significantly ($p < 0.0001$) increased compared to those of control (5.1 ± 0.4 nN, 19.4 ± 3.0 MPa, respectively) (Fig 2C and 2D).

## Membrane disruption with RM EO(C)s exposure

*C. albicans* RSY150 and ATCC10231 exposed to RM oil and its components at MIC and 1/2 MIC (Fig 3A and 3B and S4C Fig), the components at FICI, or the positive control Amp B at MIC showed a significant ($p < 0.0001$) increase in Dis-C2(3) fluorescence intensity, two-fold that of control, representing dissipated membrane potential.

*C. albicans* RBY1132 cells treated with EO(C)s at MIC and 1/2 MIC for 4 h (Fig 3C and 3D) had a significant ($p < 0.01$) increase in PI uptake. Amp B at 1/2 MIC and MIC also significantly ($p < 0.05$ and $p < 0.01$, respectively) increased PI uptake, less than that of EO(C) treated cells, but similar to those treated with the two components at 1/2 FICI. $H_2O_2$ at MIC, expected

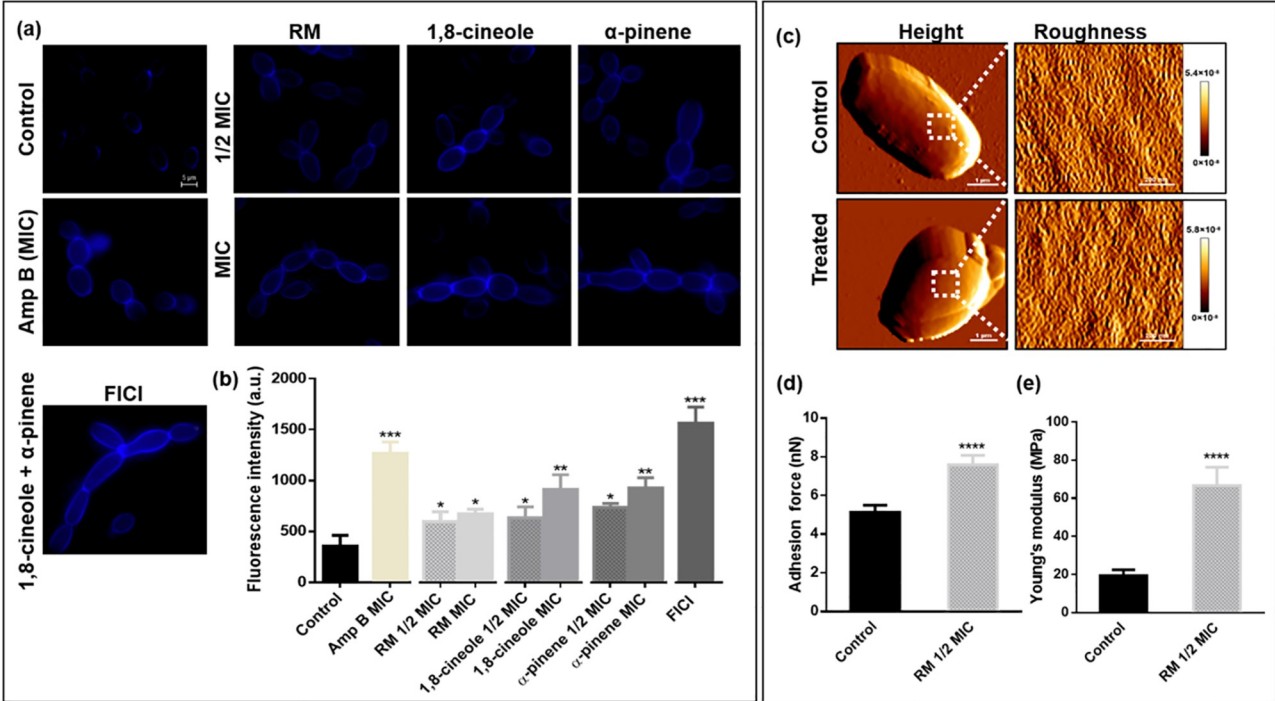

**Fig 2. Effects of RM oil, 1,8-cineole, and α-pinene on *C. albicans* RSY150 cell separation, chitin production and surface remodelling. (a)** Representative epifluorescence images of *C. albicans* with our without exposure to oils for 4h, followed by calcofluor white staining. Cells in media served as a negative control. Scale bar for the control is 5 μm and represents the scale for all images. **(b)** Bar graphs of total fluorescence intensity. AFM QI™ data for control and RM oil treated *C. albicans* at **(c, left)** low (10 μm, 128 × 128 pixel) **(c, right)** and high (1 μm, 128 × 128 pixel) resolution was used to generate representative topographic images of the cell surface (boxed area images), revealing no change in surface roughness (visually and quantitatively), but the data indicate a significant increase in **(d)** adhesion and **(e)** elasticity for treated cells. Data are presented as the mean ± SEM of three biological replicates, with **(b)** 300 representative cells per replicate, or **(d and e)** 17 cells per replicate, for which statistical significance (****, $p < 0.0001$; ***, $p < 0.001$; **, $p < 0.01$; *, $p < 0.05$) was analysed by an unpaired Student's *t*-test of treated compared to control.

to kill cells, significantly ($p < 0.001$) increased PI uptake, 24-fold higher than untreated controls and 2–1.8-fold higher than EO(C)-treated samples.

## RM EO(C)s elicit vacuole stress response and disrupt mitochondrial function

Since osmotic and drug induced cell stress are known to cause vacuolar segregation [71], we examined the impact of RM oil and its components on vacuoles. Bright field images of control RSY150 cells show the majority of cells with large, clearly visible vacuoles, occupying approximately half of the cell volume. In contrast, mid logarithmic phase *C. albicans* RSY150 and ATCC10231 exposed to RM oil or its components at MIC or 1/2 MIC, or the combined components at FICI had a significantly ($p < 0.001$ and $p < 0.01$, respectively) increased number of segregated vacuoles (Fig 4A and 4B). Interestingly, the pseudohyphal cells that formed following exposure had similar vacuolar effects. Vacuole disruption was also observed during Amp B-induced cell death at MIC (S4D Fig).

Mitochondrial membrane integrity was assessed with Mitotracker deep red staining of *C. albicans* RBY1132 following 4 h exposure to EO(C)s, showing untreated cells with red, dense mitochondria throughout, which were not visible following exposure to EO(C)s (Fig 4C and 4D).

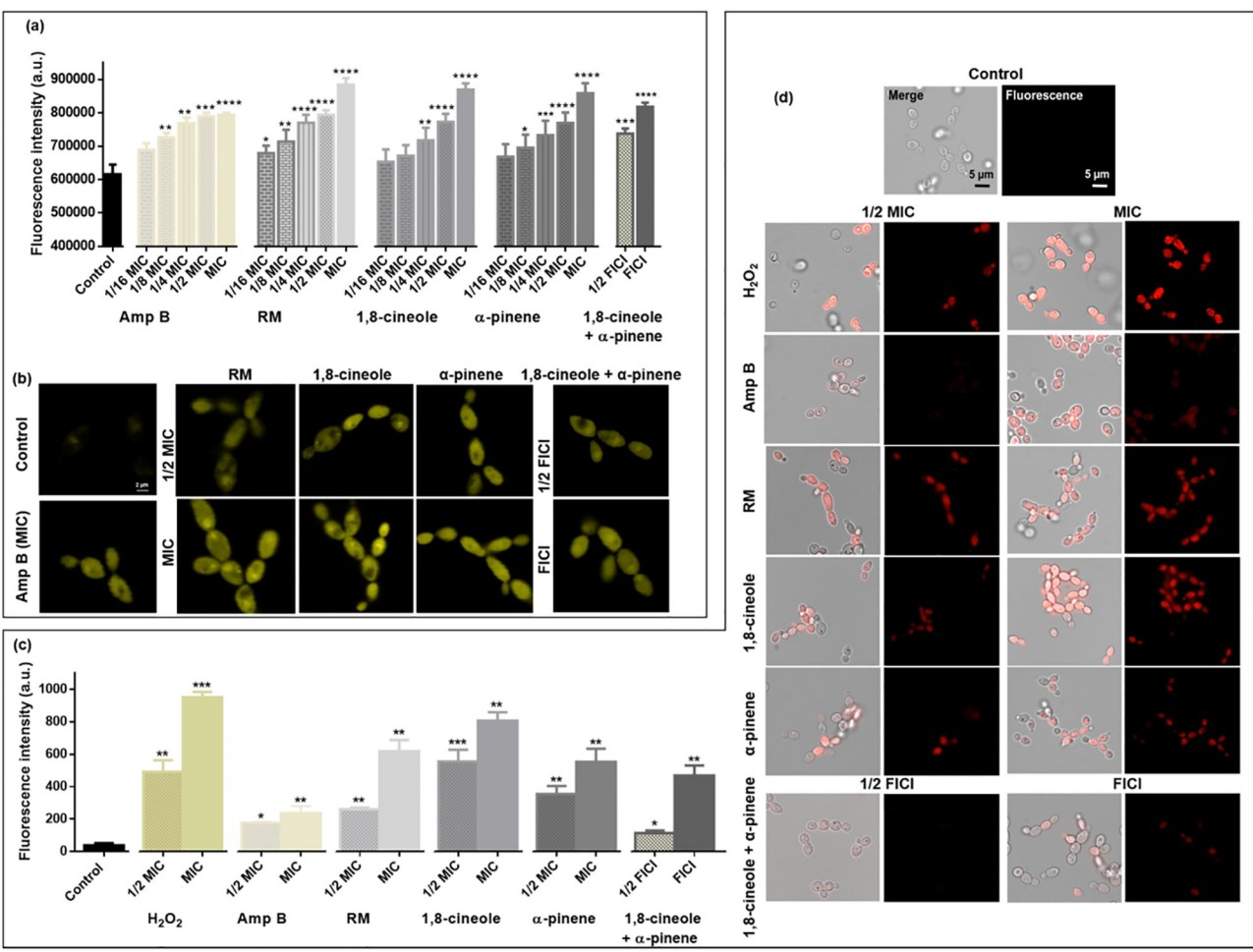

**Fig 3. Effects of RM oil, 1,8-cineole and α-pinene on *C. albicans* RSY150 cell membrane polarization and on *C. albicans* RBY1132 cell membrane integrity. (a)** Bar graphs show a significant (****, $p < 0.0001$; ***, $p < 0.001$; **, $p < 0.01$; *, $p < 0.05$) increase in *C. albicans* RSY150 membrane depolarization with exposure to EO(C)s at MIC, 1/2 MIC and the two components at FICI, compared to the controls. Pearson correlation indicates a positive association ($r = 0.96–0.99$, $p < 0.001–0.01$) between the oil concentration (1/16, 1/8, 1/4, 1/2 MIC and MIC) and membrane depolarization. **(b)** LSCM images show increased dye content in treated cells compared to control. Scale bar for control is 2 μm and represents that for all images. **(c)** Bar graphs show significantly (***, $p < 0.001$; **, $p < 0.01$ *, $p < 0.05$) higher PI uptake in oil treated *C. albicans* RBY1132 compared to control, with 100 cells per replicate. **(d)** Representative merged (bright-field/fluorescence) **(left)** and fluorescence **(right)** images ($\lambda_{ex} = 493$ nm; $\lambda_{em} = 636$ nm) of treated *Candida* show PI uptake compared to controls. Scale bars are 5 μm, applicable to all images. Data in bar graphs are presented as the mean ± SEM of three biological replicates, as evaluated by **(a)** a one-way ANOVA, followed by Dunnett's multiple comparison of each condition versus control or **(c)** by an unpaired Student's *t*-test.

## *C. albicans* exposed to RM EO(C)s accumulate intracellular ROS

Since mitochondrial dysfunction can lead to ROS production, intracellular ROS was assessed using the redox sensitive fluorescent dye, DCFDA in *C. albicans* RBY1132 exposed to EO(C)s. Cells exposed to the peroxide control, RM oil and its components at MIC and 1/2 MIC had significant (RM oil, 1,8-cineole and $H_2O_2$; $p < 0.0001$; α-pinene; $p < 0.05$) ROS accumulation (S5 Fig), which was absent in untreated cells or those treated at from 1/4 to 1/16 MIC (Fig 5). *C. albicans* showed elevated ROS when treated with RM oil (14-fold), 1,8-cineole (5.3-fold), α-pinene (2.2-fold) and $H_2O_2$ (7-fold) at 1/2 MIC (25 mM) compared to controls.

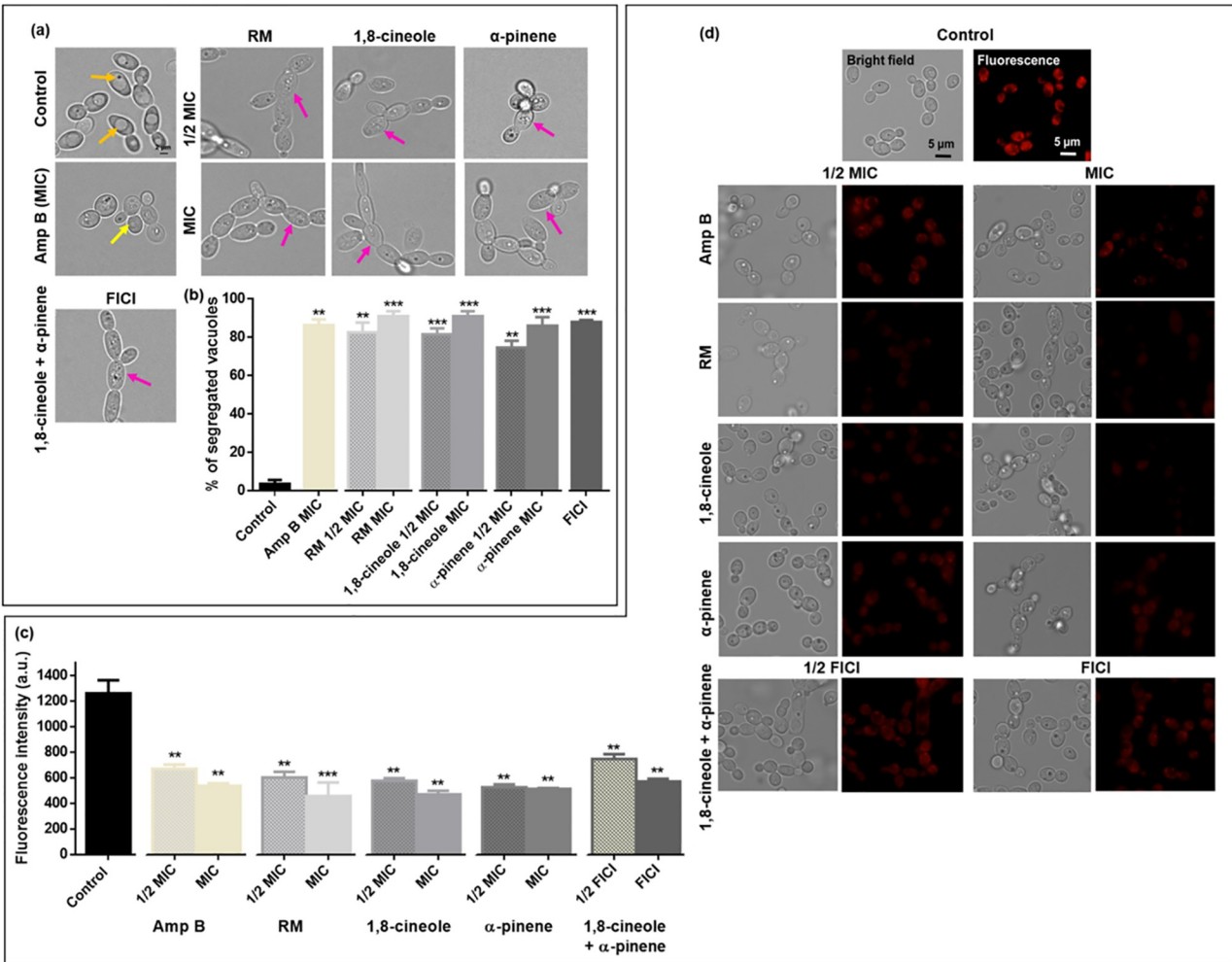

**Fig 4. Impact of RM oil and its components on *C. albicans* organelles. (a)** RM oil, 1,8-cineole, and α-pinene at MIC, and the two components at FICI, induced vacuolar fragmentation (pink arrows) as compared to control (orange arrow). The positive control, Amp B, at MIC showed partial (yellow arrow) segregation compared to EO(C)-treated cells. Scale bar for control is 2 μm and represents the scale for all images. **(b)** Bar graphs show significant differences in % segregated vacuoles for control versus treated cells. **(c)** Bar graphs show *C. albicans* exposed to EO(C)s have a greater number of damaged mitochondria compared to control. **(d)** Representative bright field **(left)** and fluorescence **(right**; λ_ex = 644 nm; λ_em = 665 nm) images of *Candida* exposed to RM oil, 1,8-cineole, α-pinene at 1/2 MIC and MIC, and the two components at 1/2 and full FICI showed poor uptake of the Mitotracker deep red as compared to control. Scale bars of control images are 5 μm and applicable to all images. Data are presented as the mean ± SEM of three biological replicates, with 300 and 100 cells per replicate for vacuole and mitochondria, respectively, for which statistical significance (***, $p < 0.001$; **, $p < 0.01$) was analysed by an unpaired Student's *t*-test.

## RM EO(C)s disrupt *C. albicans* microtubules

Since increased ROS production is known to elicit oxidative damage to nucleic acids, proteins and microtubules (MTs), disruption of vital cellular processes including cell cycle machinery was expected [72, 73]. MT length in untreated RSY150 cells varied, but consistently spanned the distance between mother and daughter cells, with normal nuclear migration (Fig 6A), in stark contrast to those exposed to RM oil at MIC, and 1,8-cineole at MIC and 1/2 MIC, which were barely visible. *C. albicans* exposed to RM oil or α-pinene at 1/2 MIC and the two major components at their FICI had only several small fluorescence spots, (Fig 6A) and interestingly, those exposed to EO(C)s at 1/4 MIC only generated beta-tubulin clusters (small fluorescent

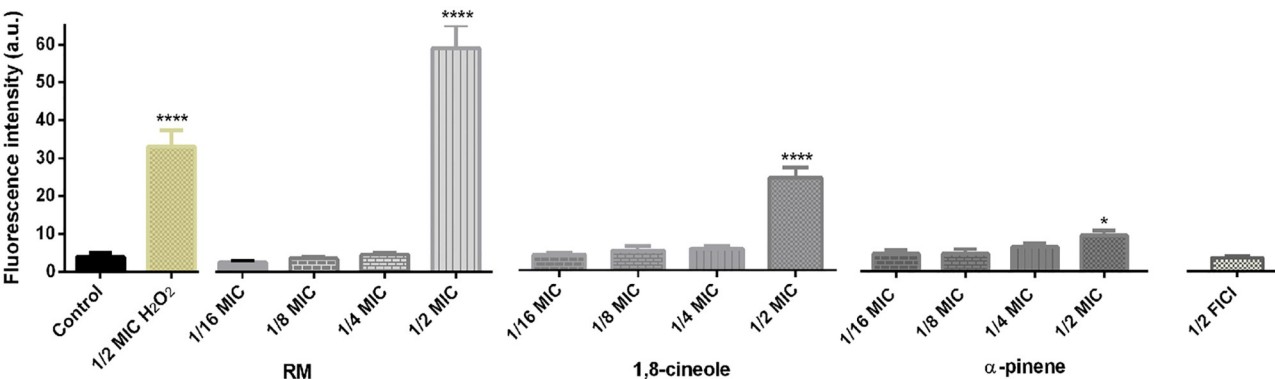

**Fig 5. Impact of RM oil and its components on *C. albicans* intracellular ROS accumulation.** Bar graphs of intracellular ROS in untreated *C. albicans* RBY1132, and those treated with RM oil, 1,8-cineole, α-pinene and $H_2O_2$ (25 mM). Fluorescence intensity was measured in a plate reader ($\lambda_{ex}$ = 485 nm; $\lambda_{em}$ = 528 nm, gain 35) and data are presented as the mean ± SEM of three biological replicates for which statistical significance (\*\*\*\*, $p < 0.0001$; \*, $p < 0.05$) was evaluated by a one-way ANOVA, followed by Dunnett's multiple comparison of each condition versus control.

spots). *C. albicans* exposed to Amp B at MIC and nocodazole at MIC and 1/2 MIC lacked MTs, consistent with EO(C)-treated cells.

Tubulin polymerization to MTs (long), aggregated tubulin (short) and diffuse beta-tubulin was quantified by counting and measuring MT length as a function of Tub2-GFP fluorescent regions [63]. MTs were considered bright fluorescent regions with a length span from 2.5 ± 0.3 to 9.4 ± 0.5 μm and those with a length range of 0.09 ± 0.01 to 2.2 ± 0.04 to μm were considered aggregated tubulin. Fluorescence intensities of the latter structures ranged from 54.8 ± 29.8 to 96.5 ± 53.3 arbitrary units (AU), whereas diffuse beta-tubulin (unpolymerized) was dispersed throughout the cell, with intensities ranging from 23.5 ± 3.8 to 41.6 ± 9.6 AU.

Cells treated with EO(C)s at 1/4 (Fig 6B) and 1/2 (Fig 6C) MIC had a statistically ($p < 0.01$ and $p < 0.001$, respectively) greater abundance of aggregated tubulin in contrast to those treated at MIC (Fig 6D) which had a significantly ($p < 0.001$) greater number of cells with diffuse beta-tubulin. The positive control, nocodazole at MIC, also had a significant increase ($p < 0.05$) of aggregated tubulin. Overall, the results showed defects in MT formation when treated at any concentration of EO(C).

## Cell cycle arrest at G1/S phase with RM EO(C) exposure

Cell cycle arrest is expected to accompany microtubular dysfunction, so LSCM was used to visualize nuclear division and MT position in the RSY 150 strains containing Tub2-GFP and Htb-RFP. Yeast cells with a single nucleus and no bud were defined as G1 phase, those with a small bud not having entered mitosis as S phase, those with nuclei spanning the mother and daughter cells as mitotic anaphase (G2), and those showing separate nuclei in both parent and budding cells were considered to be post-mitotic, M phase [63]. RSY 150 cells exposed to RM oil, 1,8-cineole, α-pinene at MIC and the latter at FICI were significantly delayed at the G1/S phase, with 70% ($p < 0.0001$); 65% ($p < 0.001$), 56% ($p < 0.01$) and 57% ($p < 0.05$), respectively, compared to untreated controls (45%) (Fig 6C). When exposed to RM oil at 1/2 MIC, 1,8-cineole at MIC and 1/2 MIC and α-pinene at MIC, there was a two-fold increase in the G2 phase compared to controls, which indicates secondary arrest at this phase. Expectedly, there was a significant ($p < 0.0001$ and $p < 0.001$, respectively) decrease in M phase cells following exposure to RM oil at MIC and 1/2 MIC, 1,8-cineole ($p < 0.0001$ and $p < 0.01$, respectively), α- pinene at MIC ($p < 0.001$) and the EO(C)s at their FICI ($p < 0.01$).

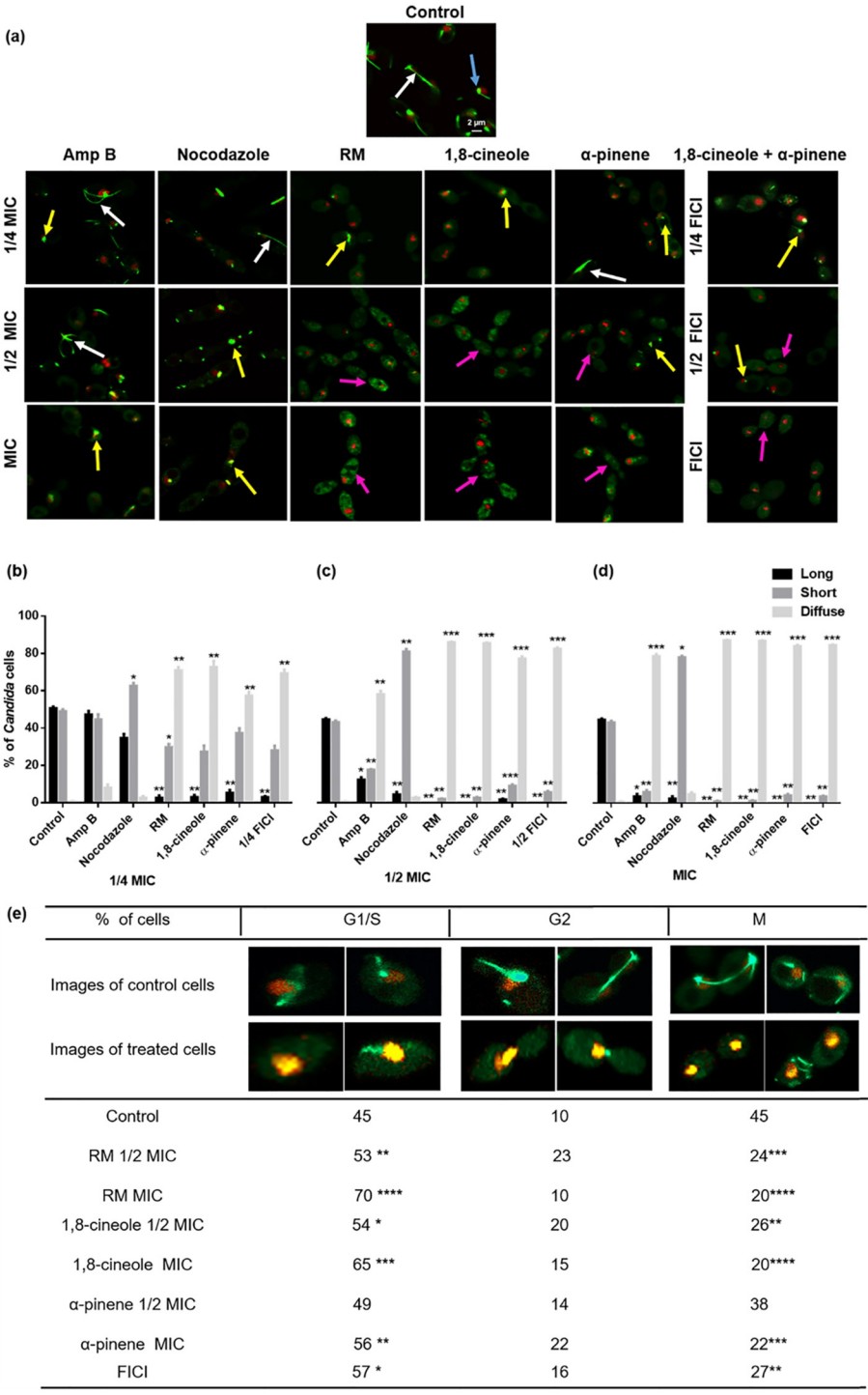

**Fig 6. Effects of RM oil, 1,8-cineole, and α-pinene on *C. albicans* RSY150 microtubule formation and cell cycle phases. (a)** The majority of cells (90–95%) treated with RM oil and its components at 1/2 MIC and MIC show diffuse tubulin (purple arrows), compared to untreated controls with normal spindle formation during mitosis (long MTs (white arrows) and short (blue arrows) MTs. Cells exposed to the positive control, nocodazole at 1/2 MIC and MIC had concentrated spots of green fluorescence (yellow arrow) similar to EO(C)-treated cells at 1/4 MIC. Images were collected by LSCM (Tub2-GFP $\lambda_{ex}$ = 488 nm; $\lambda_{em}$ = 512 nm and Htb-RFP $\lambda_{ex}$ = 543 nm; $\lambda_{em}$ = 605 nm). Scale bar for control is 2 μm and represents the scale for all images. **(b–d)** Bar graphs of β-tubulin arrangement (long MT, short MT, diffuse). **(e)** Cell cycle (S/G1, G2, or M phase) frequency (%) following exposure to RM oil, 1,8-cineole, α-pinene at 1/2 MIC and MIC and the two components at FICI show arrest at the G1/S phase. Data are presented as the mean ± SEM

of three biological replicates, with 250 cells per replicate, for which statistical significance ((****, $p < 0.0001$; ***, $p < 0.001$; **, $p < 0.01$; *, $p < 0.05$), was evaluated by **(b–d)** a one-way ANOVA, followed by Dunnett's multiple comparison of each condition versus control for or **(e)** an unpaired Student's *t*-test.

**(ii) ROS-independent anti-virulence activity of EO(C)s at low fractional MIC.** At lower EO(C) exposure levels (1/8–1/4 MIC) *C. albicans* virulence factors were inhibited in a ROS-independent manner.

### RM EO(C)s inhibit *C. albicans* germ tube formation

The yeast to hyphal transition of RSY150 and ATCC10231, induced by the presence of 10% serum, was inhibited when cells were exposed to RM oil and its components, with a significant ($p < 0.0001$) reduction (75 to 90%) in the number of germ tubes, depending on the oil type and concentration (Fig 7A and 7B and S6A Fig). Clinical isolates (Cli-1, Cli-2, Cli-3) exposed to the essential oils had fewer, shorter and swollen germ tubes compared to RSY150 and ATCC10231 (S6B–S6D Fig), and so more resistance to germ tube inhibition. The essential oils at MIC, and the major component, 1,8-cineole, were the most effective in inhibiting germ tube formation (S5 Table). Interestingly, following 4 h treatment of RSY150 with the two components at FICI (1/2 MIC 1,8-cineole + 1/8 MIC α-pinene), the number of germ tube forming cells are higher as compared to those exposed to individual EOCs at sub MIC (S7 Fig). Therefore, the kinetics of germ tube formation was used to identify the time point when there was 90% reduction, showing a statistically significant ($p < 0.0001$) reduction at the 12 h mark (S7B Fig), comparable to that exposed for 4 h to either component at MIC (Fig 7B).

### RM EO(C)s reduce preformed *C. albicans* biofilm

Stereoscopic bright field images of control RSY150 biofilms show a multilayer of dense filamentous cells (S8 Fig), in contrast to the absence of filamentous cells for those exposed for 24 h with EO(C)s at MIC. The MTT assay of *C. albicans* RSY150 exposed to RM oil and its components shows a significant ($p < 0.0001$) dose dependent (r = 0.95 – 0.98) reduction in biofilm formation (Fig 7C). There was only a modest reduction in biofilm formation with exposure to EO(C)s at 1/8 MIC ($p < 0.01$) and no significant difference for 1/16 MIC, with the exception of 1,8-cineole ($p < 0.05$). As expected, exposure to Amp B at MIC and 1/2 MIC also significantly ($p < 0.0001$) reduced biofilm formation.

### RM EO(C)s hinder *C. albicans* morphological switching

With the constant presence of various concentrations (1/2 MIC to 1/16 MIC) of RM oil, 1,8-cineole, α-pinene and the latter two at their 1/2 FICI, there was reduced mycelial formation by *C. albicans* RSY150. When grown on spider media, colonies of treated cells appeared smooth and round, whereas the colonies on the control plate were highly wrinkled and uneven (Fig 7D), supporting the data on hyphal inhibition (Fig 7A and 7B). Control colonies, covered with dense masses of mycelium, had hyphal filaments (mycelial growth) at their edges, similar to those arising from *C. albicans* exposed to each oil at 1/16 MIC, with slightly less prominent mycelia compared to control. This is in stark contrast to colonies of *C. albicans* exposed to 1,8-cineole and α-pinene at 1/2 MIC and the two at 1/2 FICI, which had no mycelia, but

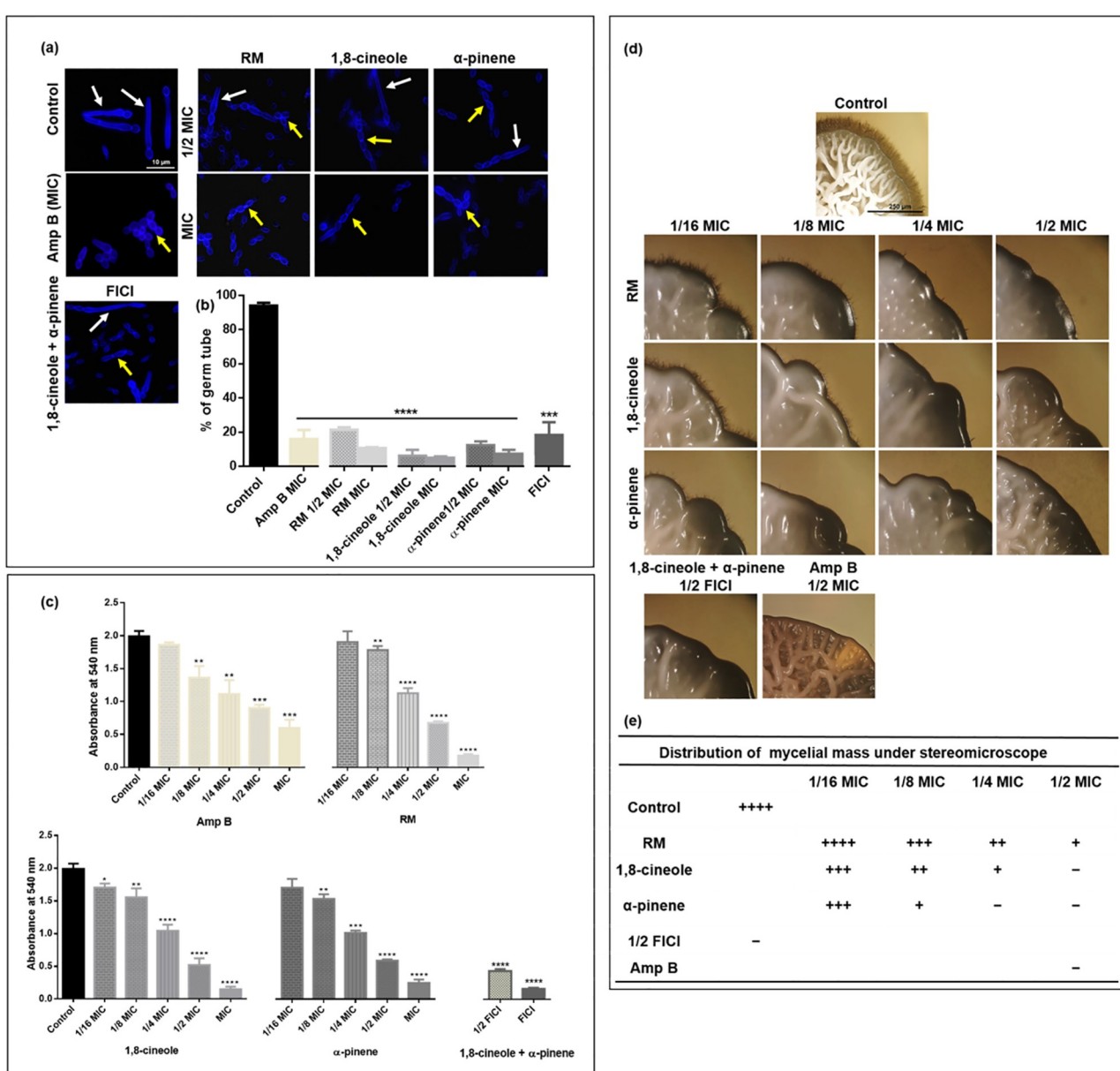

**Fig 7. RM oil and its components impact *C. albicans* RSY150 germ tube, mycelium formation and preformed biofilm. (a)** Representative fluorescence microscopy ($\lambda_{ex}$ = 365 nm; $\lambda_{em}$ = 435 nm) images show that *C. albicans* treated with 10% FBS in YPD medium containing either RM oil, 1,8-cineole or α-pinene at 1/2 MIC or MIC, or the latter at their FICI (4 h exposure) form pseudohyphae (yellow arrows), while control cells mostly produced germ tubes (white arrows). Scale bar for the control is 10 μm and applies to all images. **(b)** Bar graphs show all EO(C)s inhibit the yeast to hyphal transition in *C. albicans*. **(c)** Bar diagram of the MTT assay shows a dose dependent (r = 0.95–0.99, $p < 0.001$ for all conditions, FICI: r = 0.86, $p < 0.05$) biofilm reduction after treatment with RM oil, 1,8-cineole, and α-pinene. Data are presented as the mean ± SEM of three biological replicates, with 100 cells per replicate, for which statistical significance (****, $p < 0.0001$; ***, $p < 0.001$; **, $p < 0.01$; *, $p < 0.05$) was analysed by **(b)** an unpaired Student's *t*-test and **(c)** a one-way ANOVA, followed by Dunnett's multiple comparison of each condition versus control. **(d)** Representative stereoscopic bright-field images of control and treated RSY150 on spider media. Bar for control is 250 μm, and represents the scale for all images. **(e)** Results are summarized in tabular format for which ++++, +++, ++ and + indicate the relative amount of mycelial growth.

slightly wrinkled colonies (Fig 7D). In all cases, cells exposed to α-pinene at 1/4 and 1/8 MIC had few or no hyphae, representing greater inhibition of morphological switching than RM oil and 1,8-cineole at those concentrations (Fig 7E). Representative images of the effect of EO(C)s on whole colonies are shown in S9 Fig.

**Table 1. Summary of mutant strain sensitivity.**

| Name of EO(C)s → Name of strains ↓ | RM (µg/ml) | 1,8-cineole (µg/ml) | α-pinene (µg/ml) | Amp B (µg/ml) |
|---|---|---|---|---|
| *als1Δ/Δ* | Sensitive | Sensitive | Highly sensitive | No change |
| *als3Δ/Δ* | No change | No change | Sensitive | No change |
| *hwp1Δ/HWP1⁺* | Sensitive | Sensitive | Highly sensitive | No change |
| *efg1Δ/Δ* | Sensitive | Sensitive | Highly sensitive | Sensitive |

## *C. albicans* strains defective in virulence genes are susceptible to RM EO(C)s

Transcription factor EFG1 and adhesion-related genes *ALS1*, *ALS3* and *HWP1* are known to contribute to *C. albicans* virulence [74], Susceptibility tests show *als1Δ/Δ*, *hwp1Δ/HWP1⁺*, and *efg1Δ/Δ* are more sensitive to EO(C)s than the background SC5314 strain (S10A–S10D Fig), and the effect is most prominent for α-pinene. The strain lacking the Als3 protein (*als3Δ/Δ*), involved in yeast-to-host tissue adhesion and yeast aggregation, was only sensitive to α-pinene. Interestingly, only *efg1Δ/Δ* was susceptible to Amp B (Table 1).

## *C. albicans* pretreated with RM EO(C)s were slower growing with less biofilm

Initial assays of mycelial growth examined planktonic *C. albicans* and preformed candidal bio-films under constant exposure to various concentrations of RM oil, 1,8-cineole, and α-pinene (MIC to 1/16 MIC), so it was of interest to test the lingering impact of a 4 h pre-exposure to EO(C)s at 1/2 MIC. Pretreated RSY150, ATCC10231 and Cli-1 had no visible growth on spider media after one day exposure to 1,8-cineole at 1/2 MIC, nor did ATCC10231 or Cli-1 exposed to RM oil at 1/2 MIC, as compared to the negative control and Amp B at 1/2 MIC (S11A Fig), demonstrating that pre-treated *Candida* grew more slowly than control at day 3. Of note, RSY150 and ATCC10231 exposed to 1,8-cineole had minimal growth at day 6.

All *Candida* isolates pretreated with EO(C)s were unable to produce biofilm 24 h later, as quantified by crystal violet staining, except the Cli-1 strain (S11B Fig). Overall, the clinical strains were more resistant than RSY150 and ATCC10231 to the EO(C)s at 1/2 MIC (S6 Table). Interestingly, the Cli-1 biofilm was comparable to control under the same conditions, which is consistent with the germ tube inhibition assay (S6B Fig), indicating more resistance compared to the other clinical strains. In contrast, 1,8-cineole and Amp B inhibited Cli-2 bio-film formation.

## Discussion

Previous studies have reported that EO(C)s can exert anti-candida activity by altering cell membrane permeability [75, 76], reducing ergosterol biosynthesis [77], and causing mitochon-drial membrane damage [78], which then leads to ROS [79], ion transport imbalance, inhibi-tion of protein function [80] and ultimately cell death. In this study we demonstrate how RM and its components, 1,8-cineole and α-pinene, not only induce ROS-dependent cell death at higher concentrations, but for the first time their ability to modulate key virulence factors in a ROS-independent manner at lower fractional MICs.

*C. albicans* exposed to RM oil, 1,8-cineole and α-pinene exhibited pseudohyphal and clus-ter-forming cells, with elevated chitin levels (Fig 2 and S4A and S4B Fig) and stiffness consis-tent with studies on classical antifungals [81–84]. It is well-known that when yeast cells experience parietal stress, the first line of defense is to overproduce chitin to reinforce the cell

wall [85], altering its nanomechanical properties [81–83]. Thus, RM-induced cell wall stiffening can be attributed to increased *C. albicans* cell wall chitin content, whereas the increased surface adhesion to the hydrophilic AFM tip (Fig 3B) is most likely attributed to cell wall remodelling [81, 82].

Smaller monoterpenes/monoterpenoids are known to infiltrate and damage cell membranes [77], consistent with the RM-induced increase in membrane permeability (Fig 3 and S4C Fig) and gene products conferring sensitivity and resistance being localized to the cell membrane (S3 Fig). However, membrane damage is not limited to the plasma membrane, and as EO(C)s infiltrate the cell its organelle membranes are also impacted. In *C. albicans*, the vacuole plays a pivotal role in ion homeostasis, protein turnover [71, 86], and fungal pathogenicity [87, 88], such as vacuolar size thresholds which are required for cell cycle progression during hyphal growth [88], making them a focus in understanding antifungal toxicity mechanisms in *C. albicans*. Vacuolar defects were apparent with RM oil, 1,8-cineole, α-pinene exposure along with the known membrane disruptor, Amp B [89], at higher concentrations (Fig 4A and S4D Fig), making a link between cell cycle arrest and hyphal growth [88]. Mitochondrial membranes are crucial for cellular metabolism and energy production, so their perturbation constitutes an early, irreversible step of programmed cell death [90, 91]. This effect was observed as mitochondrial membrane potential perturbation (Fig 4B), ROS generation (Fig 5) and ultimately cell death (Fig 3) [90], similar to the effect of carvacrol [92], α- and β-pinene [19, 93]. Impact on more than one *C. albicans* organelle, supported by the chemogenomic cluster analysis (Fig 1), is consistent with a nonspecific stress response resulting from ROS overproduction, as observed for the classical antifungals, Amp B and itraconazole [94, 95].

ROS have been shown to oxidize the sulfhydryl groups of the tubulin cysteine residues [96, 97], promoting α and β-tubulin crosslinks that prevent MT assembly [97, 98], as observed for RM (Fig 6A–6D) and cinnamon bark [47] EO(C)s. MT-disrupting agents are thought to arrest the cell cycle by triggering the mitotic checkpoint. For example, drugs such as methyl benzimidazol-2-ylcarbamate, nocodazole and griseofulvin interfere with cytoskeletal assembly, eliminating astral MTs [99] and inhibiting nuclear migration and mitosis in hyphae [100], which leads to cell cycle alteration and ultimately apoptosis [101–103]. The majority of *C. albicans* exposed to 1,8-cineole and α-pinene at high concentrations arrested in the G1/S phase (Fig 6E), similar to other EO(C)s [18, 104]. Recent reports link cell wall and membrane perturbation [105, 106] to an acute block in vacuole biogenesis, resulting in the loss of functional vacuoles and a specific arrest of cells at the early G1 phase [107]. Interestingly, the strain lacking *PDS5* (Fig 1), encoded during cell-cycle regulated periodic mRNA expression and predicted to have a role in the establishment and maintenance of sister chromatid condensation and cohesion, is sensitive to both EOCs as it is for Amp B [23]. Taken together this links plasma membrane depolarization and cell cycle arrest at the G1 phase. Thus, the EO(C)s induce membrane depolarization, vacuolar defects, and ROS-induced MT dysfunction, important factors triggering G1/S cycle arrest (Fig 8A) and preventing cells from entering the G2 and M phases and limiting further growth.

Importantly, at lower fractional MICs and 1/2 FICI, there was a ROS-independent (Figs 3B and 5) inhibition of several virulence factors. Hyphal formation is a morphogenetic process that contributes to the virulence and pathogenicity of *C. albicans* [108], and strains with defects in this process produce abnormal biofilms [109, 110]. The inhibition of hyphal formation by RM EO(C)s (Fig 7A and S6 and S7 Figs) is consistent with previous data on EOs [111–113] and azoles [114]. Vacuolar biogenesis [115, 116] and spindle dynamics (Tub2) are tightly coordinated with morphology [63], which regulate hyphal growth [99, 117]. Since ROS are not generated at lower fractional MICs, vacuolar and MT dysfunction appear sufficient to inhibit hyphal activity (Fig 8B), and partial synergy of the two components appears attributable to MT

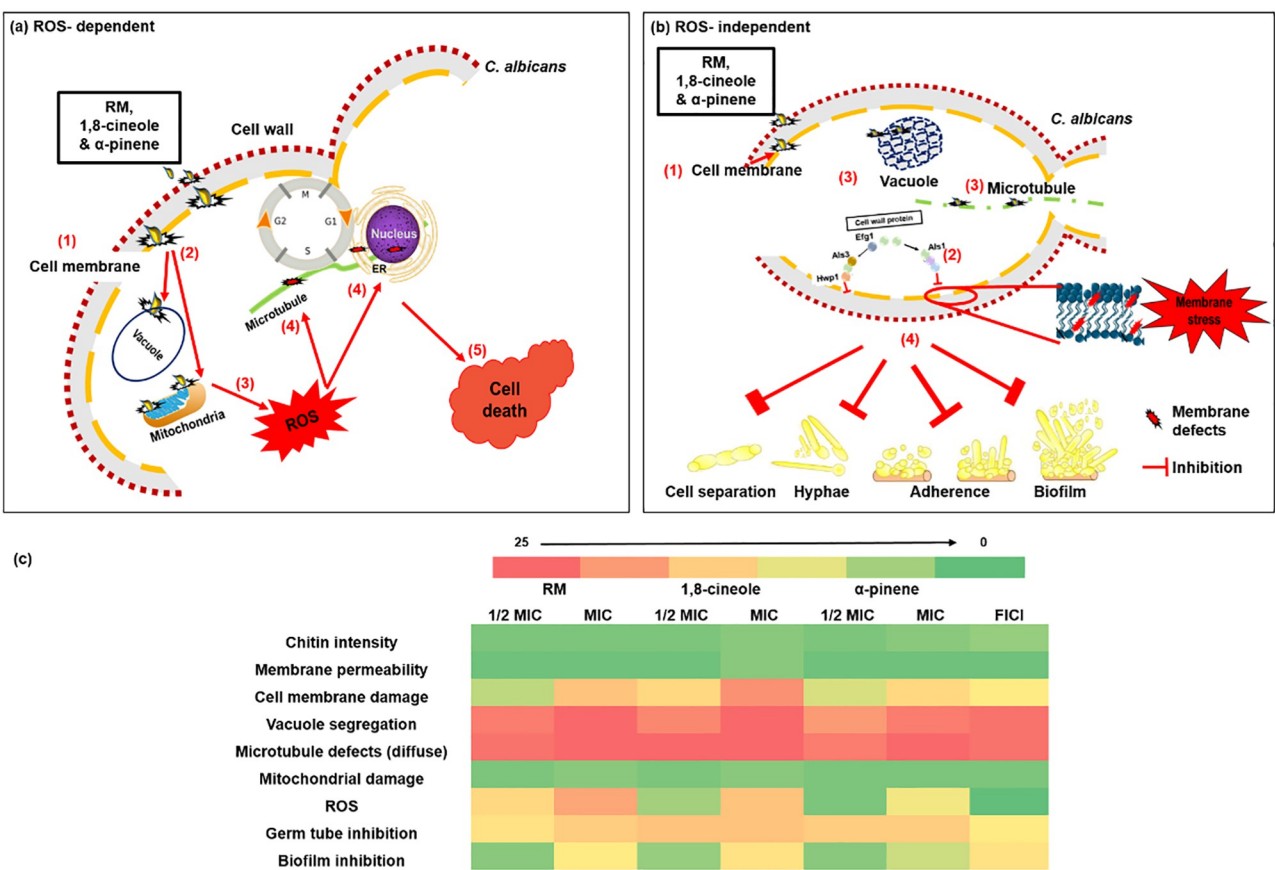

**Fig 8. Cartoon representation of potential *C. albicans* biological pathways and overall impact by RM oil and its major components.** (a) At higher concentrations (1/2 to full MIC), EOCs can pass through the *C. albicans* cell wall and gain access to (1) the cell membrane, wherein they cause membrane depolarization. (2) Once EOCs pass through the cell membrane they can enter the membrane of other organelles, such as vacuoles, mitochondria and the nucleus. (3) Mitochondria are the main source of cellular ROS, in which membrane damage would lead to cellular stress and accumulation of ROS, which subsequently (4) causes damage to macromolecules (e.g. MTs) and (5) eventually cell death. (b) At lower fractional MIC, EO(C)s also diffuse through the cell wall and (1) enter through the cell membrane, where they cause (2) membrane defects, proposed to retard major cell wall virulence proteins from passing through and reaching the cell wall surface. This would impact important cell wall virulence proteins, including Als1, Als3, Hwp1, which play an important role in adhesion, biofilm and hyphal formation. (3) Direct impacts of EO(C)s on vacuoles and MTs also play an important role in (4) inhibiting hyphal formation. Abnormal MT morphology can also contribute to the cell separation defects, which lead to pseudohyphal growth. (c) Heat map of major impacts to *C. albicans* RSY150 treated with RM oil and its components, 1,8-cineole and α–pinene. Colour scale indicates fold change in parameter with EO(C) exposure. The major *C. albicans* impacts associated with RM oil are attributable to 1,8-cineole, which accounts 53% of the essential oil.

disruption (Fig 8C) which could be a strong indication that each one acts on a different target, thus inhibiting different biochemical processes in the fungal cell [50], which requires further study.

Adhesion [118] and hyphal formation is critical for biofilm biogenesis in *Candida*, providing strength and support for the developing heterogeneous structure [119]. RM and its components effectively inhibited approximately 90% of biofilm formation (Fig 7 and S8 and S9 Figs), expected based on the antibiofilm activity of 1,8-cineole and α- pinene [120] and in line with the anti-adherent activity of RM oil [121]. Since the cell membrane plays a critical role in virulence, by mediating the secretion of virulence factors, orchestrating endocytosis, cell wall synthesis and invasive hyphal morphogenesis [122], it is plausible that even relatively subtle changes to the cell membrane structural integrity detrimentally impacts the secretion of virulence proteins. Indeed, many terpenes have anti-biofilm activity in *Candida* [120, 123, 124],

speculated to repress genes important in biofilm formation, such as *ALS1*, *ALS3*, and *HWP1* [125].

Certainly, these biofilm regulatory proteins confer *C. albicans* tolerance to RM oil, 1,8-cineole, and in particular α-pinene (S10 Fig) which also inhibits mycelial growth more effectively than RM oil and 1,8-cineole (Fig 7E). In comparison, only *EFG1* confers tolerance to Amp B, in agreement with a previous study [126]. We speculate that cell membrane damage interferes with the synthesis or transport of these proteins, thereby reducing invasion and adhesion (S10B Fig).

## Conclusions

RM oil, 1,8- cineole and α-pinene at high concentrations damage *C. albicans* cellular and organelle membranes (vacuolar and mitochondrial), leading to complete vacuolar segregation, mitochondrial dysfunction, reactive oxygen species, and microtubular aberrations, cell cycle arrest in the G1/S phase, and ultimately cell death, consistent with chemogenomic profiles that indicate a general stress response at higher concentrations. The partial synergistic effect of 1,8-cineole and α-pinene enhances RM oil impact against *C. albicans*. The ROS-independent inhibition of virulence traits at lower fractional MICs, including hyphal morphogenesis and biofilm formation, accompanied by MT dysfunction are entirely novel findings. The inhibition of fungal growth and virulence factors are viable alternatives for treating fungal infections, in particular in combination with classical antifungals, and is a promising avenue for developing novel sanitizing formulations and medical coatings aimed at preventing infection.

## Supporting information

**S1 Fig. Gas chromatogram of RM oil.**
(DOCX)

**S2 Fig. MIC of various *C. albicans* strains and EOC synergy.**
(DOCX)

**S3 Fig. *C. albicans* genes conferring resistance and sensitivity to EOCs.**
(DOCX)

**S4 Fig. Effects of RM oil and its major components 1,8-cineole and α-pinene on *C. albicans* ATCC10231.**
(DOCX)

**S5 Fig. Impact of RM and its components, 1,8-cineole and α-pinene, on *C. albicans* RBY1132 intracellular ROS accumulation.**
(DOCX)

**S6 Fig. Effects of RM, 1,8-cineole and α-pinene on germ tube formation by *C. albicans* ATCC10231 and clinical strains (Cli-1, Cli-2 and Cli-3).**
(DOCX)

**S7 Fig. Kinetics of *C. albicans* RSY150 germ tube inhibition by 1,8-cineole and α- pinene at FICI.**
(DOCX)

**S8 Fig. RM oil and its major components reduced biofilm formation in *C. albicans* RSY150.**
(DOCX)

**S9 Fig. Impact of RM oil, 1,8-cineole, and α-pinene on *C. albicans* RSY150 mycelium formation on spider media agar plates (whole colony).**
(DOCX)

**S10 Fig. Sensitivity profile of *C. albicans* als1Δ/Δ, als3Δ/Δ, hwp1Δ/HWP1⁺ and efg1Δ/Δ to EO(C)s.**
(DOCX)

**S11 Fig. Impact of pre-treating *C. albicans* RSY150, ATCC10231 and clinical isolates (Cli-1, 2 and 3) with RM, 1,8-cineole and α-pinene at 1/2 MIC on hyphal and biofilm formation.**
(DOCX)

**S1 Table. *C. albicans* strains used in this study.**
(DOCX)

**S2 Table. Concentrations of essential oil stock and working solutions used in this study.**
(DOCX)

**S3 Table. Composition of rosemary (*Rosmarinus officinalis*) essential oil from whole plants analyzed by GC/MS.**
(DOCX)

**S4 Table. CDC gene ontology functional analysis of *C. albicans* genes conferring significant EO(C)-resistance and -sensitivity.**
(DOCX)

**S5 Table. Relative germ tube inhibition by RM oil and its components at MIC.**
(DOCX)

**S6 Table. Relative reduction in biofilm formation following pre-treatment with RM oil and its components at 1/2 MIC.**
(DOCX)

## Acknowledgments

We are grateful to Dr. Richard J. Bennett, Department of Molecular Microbiology and Immunology, Brown University, USA; Dr. Lois L. Hoyer, Department of Veterinary Pathobiology, University of Illinois at Urbana-Champaign, Urbana, USA and Dr. Jessica Minion, Regina Qu'Appelle Health Region, Regina, SK Canada for providing the RSY, Als mutant and clinical strains, respectively, for this study. We thank Rebecca Jameson for the RM oil GC-FID and GC–MS analysis. This research was partially conducted on the traditional territories of the nêhiyawak, Anihsinapek, Nakoda, Dakota, and Lakota peoples, and the homeland of the Métis/Michif Nation.

## Author Contributions

**Conceptualization:** Zinnat Shahina, Taranum Sultana, Tanya E. S. Dahms.

**Data curation:** Zinnat Shahina, Raymond Al Homsi, Jared D. W. Price, Taranum Sultana.

**Formal analysis:** Zinnat Shahina, Raymond Al Homsi, Jared D. W. Price, Taranum Sultana.

**Funding acquisition:** Malcolm Whiteway, Taranum Sultana, Tanya E. S. Dahms.

**Investigation:** Zinnat Shahina.

**Methodology:** Zinnat Shahina, Raymond Al Homsi, Jared D. W. Price, Taranum Sultana, Tanya E. S. Dahms.

**Project administration:** Tanya E. S. Dahms.

**Resources:** Malcolm Whiteway, Tanya E. S. Dahms.

**Supervision:** Malcolm Whiteway, Taranum Sultana, Tanya E. S. Dahms.

**Writing – original draft:** Zinnat Shahina.

**Writing – review & editing:** Malcolm Whiteway, Taranum Sultana, Tanya E. S. Dahms.

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
