## [Decision Letter · Decision Letter 0]

29 Jul 2022

PONE-D-22-17200Rosemary essential oil and its components 1,8-cineole and α-pinene induce ROS-dependent cell death and inhibit Candida albicans virulence in a ROS-independent mannerPLOS ONE

Dear Dr. Dahms,

Thank you for submitting your manuscript to PLOS ONE. After careful consideration, we feel that it has merit but does not fully meet PLOS ONE’s publication criteria as it currently stands. Therefore, we invite you to submit a revised version of the manuscript that addresses the points raised during the review process.

Your manuscript has been reviewed by 2 specialists in the field and as you can see from the revisions the manuscript requires minor non experimental revisions before it is fit for publication. Please address all the points raised by the reviewer.

We look forward to receiving your revised manuscript.

Kind regards,

Roy Aziz Khalaf

Academic Editor

PLOS ONE

Journal Requirements:

This work was supported by Natural Science and Engineering Research Council Discovery Grant (NSERC DG; RGPIN-2018-06649), Saskatchewan Health Research Foundation Collaborative Innovation Development and Canada Foundation for Innovation (CFI) grants to TESD, a NSERC DG (RGPIN-2017-4799) and Canada Research Chair (950-228957) to MW. TS was partially supported by the Faculty of Science and CFI infrastructure operating fund to TESD. ZS was partially supported by the Faculty of Graduate Studies and Research at the University of Regina.

Additional Editor Comments:

Greetings

Your manuscript has been reviewed by 2 specialists in the field and as you can see from the revisions the manuscript requires minor non experimental revisions before it is fit for publication. Please address all the points raised by the reviewer.

Reviewers' comments:

Reviewer's Responses to Questions

**Comments to the Author**

1. Is the manuscript technically sound, and do the data support the conclusions?

Reviewer #1: Yes

Reviewer #2: Yes

2. Has the statistical analysis been performed appropriately and rigorously? 

Reviewer #1: Yes

Reviewer #2: Yes

3. Have the authors made all data underlying the findings in their manuscript fully available?

Reviewer #1: Yes

Reviewer #2: Yes

4. Is the manuscript presented in an intelligible fashion and written in standard English?

Reviewer #1: Yes

Reviewer #2: Yes

5. Review Comments to the Author

Reviewer #1: The study by Shahina et al. describes the potential mechanisms of action and antifungal effects of two major components of rosemary essential oil (alpha-pinene and 1,8-cineole) against Candida albicans. The authors make a detailed description of the potential mechanisms of action, in addition to a very careful and accurate choice of the proposed experimental design. It explores numerous universally accepted and recognized methodologies and presents the results clearly and accurately. All experiments are strictly controlled and the results are presented in tables and images of high technical and graphic quality. The theme is of interest to the biomedical community and relevant within medical mycology. The results expand the prerogative of using rosemary in pharmaceutical preparations with antifungal potential. I suggest publishing the study after the authors consider minor revisions:

Candida albicans ATCC10231 is resistant to several antifungals (such as fluconazole, anidulafungin, miconazole). In addition, the susceptibility profile of the clinical isolates used should be indicated in the supplementary material section. This information is important as it may perhaps help to explain the difference in susceptibility to alpha-pinene and 1,8-cinelol of isolates used in the study.

The authors report a weak synergistic interaction of alpha-pinene and 1,8-cinelol. In assays of genes linked to resistance to these two agents, very distinct patterns were observed between the compounds. This is a strong indication that each one acts on a different target, thus inhibiting different biochemical processes in the fungal cell. Inhibition of distinct metabolic pathways is one of the cornerstones of synergism between molecules and these correlations should be further explored in the Discussion section.

Reviewer #2: The manuscript provides promising outcomes on the efficacy of essential oil from rosemary on C. albicans. Although the manuscript shows excellent presentation, some comments still need to be addressed including

1. Title: need to be modified and concise. I would suggest "Rosemary essential oil and its major components 1,8-1 cineole and α-pinene induce ROS-dependent and ROS-independent inhibition of Candida albicans"

2. MIC need to be identified in the abstract, for example is it MIC100? also include SE

3. Also in the abstract, include fold difference compared to control when talking about mechanisms.

4. Please follow binomial system when talking about organism names. when plant name alone should be lower case. for example Rosmarinus officinalis, while rosmeray should be used. please also include plant family name and taxonomic name at first use in the abstract and materials.

5. Please include CAT# and company, country of all purchases and equipment

6. Add more details regrading the use of Software employed

7. Please give details about all negative controls and if they contain all components except the compound or drug. In other words DMSO and TWEEN can cause antifungal effect if not in suitable concentration.

8. Explain and mention in the manuscript the reason for using different CFU from experiment to another?

9. Please rewrite the text under the first subheading in the results. Also some metabolites should be considered major and the authors identified as minor. However they can identify that cineole and Pinene are the highest.

10. A section in the methods about the fungal strains in terms of source and way of isolation and maintenance should be included, in addition to any ethical approval should be used.

11. The concentration of the oil employed in the study is very high, which may suggest cytotoxicity. Can the author elaborate on this

12. Please revise the labels on Figure 8, I believe MIC should be instead of 1/2 MIC under pinene. also revise all labels within all figures through out the manuscript . Further, please make all letters in the figure legend bold to differentiate from text.

13. Although the English is fairly used, so many mistakes, spelling errors, incomplete sentences and sentences construction are not satisfactorily used and need to be carefully revised.

6. PLOS authors have the option to publish the peer review history of their article (what does this mean?). If published, this will include your full peer review and any attached files.

Reviewer #1: **Yes: **William Gustavo Lima

Reviewer #2: **Yes: **Sameh Soliman

---

## [Author Response · Author response to Decision Letter 0]

14 Oct 2022

We thank the reviewers for their careful review which ultimately improve our submission. All changes in the manuscript appear in blue font.

5. Review Comments to the Author

Reviewer #1: The study by Shahina et al. describes the potential mechanisms of action and antifungal effects of two major components of rosemary essential oil (alpha-pinene and 1,8-cineole) against Candida albicans. The authors make a detailed description of the potential mechanisms of action, in addition to a very careful and accurate choice of the proposed experimental design. It explores numerous universally accepted and recognized methodologies and presents the results clearly and accurately. All experiments are strictly controlled and the results are presented in tables and images of high technical and graphic quality. The theme is of interest to the biomedical community and relevant within medical mycology. The results expand the prerogative of using rosemary in pharmaceutical preparations with antifungal potential. I suggest publishing the study after the authors consider minor revisions:

Candida albicans ATCC10231 is resistant to several antifungals (such as fluconazole, anidulafungin, miconazole). In addition, the susceptibility profile of the clinical isolates used should be indicated in the supplementary material section. This information is important as it may perhaps help to explain the difference in susceptibility to alpha-pinene and 1,8-cinelol of isolates used in the study.

We have included the MIC information for each antifungal and each strain, including clinical isolates in the supplementary material, S2 Fig entitled “MIC of various C. albicans strains and EOC synergy.”. In comparison with the clinical reference strain (ATCC10231), only the genital and blood strains show resistance to �-pinene. 

The authors report a weak synergistic interaction of alpha-pinene and 1,8-cinelol. In assays of genes linked to resistance to these two agents, very distinct patterns were observed between the compounds. This is a strong indication that each one acts on a different target, thus inhibiting different biochemical processes in the fungal cell. Inhibition of distinct metabolic pathways is one of the cornerstones of synergism between molecules and these correlations should be further explored in the Discussion section.

Thank you for pointing out this oversight. Indeed there could be inhibition of distinct metabolic pathways which leads to synergism, and we have added this idea to the discussion (lines 801-803).

Reviewer #2: The manuscript provides promising outcomes on the efficacy of essential oil from rosemary on C. albicans. Although the manuscript shows excellent presentation, some comments still need to be addressed including

1. Title: need to be modified and concise. I would suggest "Rosemary essential oil and its major components 1,8-1 cineole and α-pinene induce ROS-dependent and ROS-independent inhibition of Candida albicans"

We thank you for this suggestion to improve the cumbersome title. We have accordingly changed the title to: Rosemary essential oil and its major components 1,8-cineole and α-pinene induce ROS-dependent lethality and ROS-independent virulence inhibition of Candida albicans"

We think it is important to distinguish between lethality and inhibition of virulence.

2. MIC need to be identified in the abstract, for example is it MIC100? also include SE

Yes, this is MIC100, which we have now indicated in the abstract (line 33). The MICs determined from three biological replicates were identical, which would represent a SEM of 0. We have modified S2 Fig legend to indicate the number of biological replicates, and typical error (~0.01) associated with 100% inhibition corresponding to a null OD600 reading (lines 1347-1348).

3. Also in the abstract, include fold difference compared to control when talking about mechanisms.

The fold change is dependent on oil type and strain, so in the abstract we have included the maximal fold changes for mycelial and biofilm formation (line 42).

4. Please follow binomial system when talking about organism names. when plant name alone should be lower case. for example Rosmarinus officinalis, while rosmeray should be used. please also include plant family name and taxonomic name at first use in the abstract and materials.

Thank you for pointing out this error. We have corrected the abstract (line 27) and methods (line 105) accordingly, and consistently use lower case for “rosemary” (lines 378). 

5. Please include CAT# and company, country of all purchases and equipment

We have corrected this oversight by including the CAT# and all suppliers (lines 107-115, 176-177, 202, 211-212, 268, 306, 362-363).

 

6. Add more details regrading the use of Software employed

We have added software for chemogenomic data processing (lines 185-187), AFM (215 and 220), fluorescence microscopy (lines 243, 258), ImageJ (lines 280-281) and statistical analyses (367-368).

7. Please give details about all negative controls and if they contain all components except the compound or drug. In other words DMSO and TWEEN can cause antifungal effect if not in suitable concentration.

Thank you for pointing out this oversight. We have included a description of negative controls (lines 114-116).

8. Explain and mention in the manuscript the reason for using different CFU from experiment to another?

Upon further review, we realize that the way in which the methods were written was confusing, so we have edited this section for clarity (lines 130-140). Included are the guidelines for yeast determined the CFU used for the MIC assays. The standard most commonly used in the clinical microbiology laboratory is the 0.5 McFarland Standard (1.2 x 105 CFU/mL). We now state that all other experiments started with a higher CFU/mL (107), at mid log phase of the growth curve where cells are more metabolically. Microscopic experiments required a higher cell density, and so were not further diluted prior to sample preparation, whereas each well for biochemical assays were normalized to 1 x 105 cells/mL, a cell density that prevents signal saturation. 

9. Please rewrite the text under the first subheading in the results. Also some metabolites should be considered major and the authors identified as minor. However they can identify that cineole and Pinene are the highest.

Thank you for pointing out the lack of clarity in this paragraph, which we have rewritten (lines 379- 385).

10. A section in the methods about the fungal strains in terms of source and way of isolation and maintenance should be included, in addition to any ethical approval should be used.

C. albicans strains were not collected by our laboratory, rather the Regina Qu’appelle Health Region microbiology laboratory, with all patient identifying information associated with clinical isolates removed prior to transfer to our laboratory. Therefore the study did not require ethical approval. We now indicate the source of each strain, which are described in the supplementary S1 Table, and we have clarified this at the beginning of our methods section (lines 91 -97). In that same paragraph, under Strains, media and growth conditions, we describe how the strains were maintained. 

11. The concentration of the oil employed in the study is very high, which may suggest cytotoxicity. Can the author elaborate on this?

We agree, which is why we do not propose their use as therapeutics, rather we suggest them as possible prophylactics: “a promising avenue for developing novel sanitizing formulations and medical coatings aimed at preventing infection”. It is possible that the chemical structures of these compounds could serve as a starting point for future therapeutics.

12. Please revise the labels on Figure 8, I believe MIC should be instead of 1/2 MIC under pinene. also revise all labels within all figures through out the manuscript . Further, please make all letters in the figure legend bold to differentiate from text.

We assume that the reviewer is referring to the close proximity of the legend to the labels in Figure 8, where we have now added a space. All Figure labels now appear in bold font, along with letters in the figure legends.

13. Although the English is fairly used, so many mistakes, spelling errors, incomplete sentences and sentences construction are not satisfactorily used and need to be carefully revised.

Agreed. We have tried to eliminate all typos, incomplete sentences and grammatical errors.

---

## [Editor Report · Decision Letter 1]

19 Oct 2022

Rosemary essential oil and its components 1,8-cineole and α-pinene induce ROS-dependent lethality and ROS-independent virulence inhibition in Candida albicans

PONE-D-22-17200R1

Dear Dr. Dahms,

We’re pleased to inform you that your manuscript has been judged scientifically suitable for publication and will be formally accepted for publication once it meets all outstanding technical requirements.

Kind regards,

Roy Aziz Khalaf

Academic Editor

PLOS ONE

Additional Editor Comments (optional):

all reviewer required corrections were performed
---

## [Editor Report · Acceptance letter]

7 Nov 2022

PONE-D-22-17200R1 

Rosemary essential oil and its components 1,8-cineole and α-pinene induce ROS-dependent lethality and ROS-independent virulence inhibition in *Candida albicans*

Dear Dr. Dahms:

I'm pleased to inform you that your manuscript has been deemed suitable for publication in PLOS ONE. Congratulations! Your manuscript is now with our production department. 

Kind regards, 

on behalf of

Dr. Roy Aziz Khalaf 

Academic Editor

PLOS ONE